# The lingering effects of Neanderthal introgression on human complex traits

Xinzhu Wei[1†], Christopher R Robles[2†], Ali Pazokitoroudi[3], Andrea Ganna[4,5,6], Alexander Gusev[7], Arun Durvasula[8,9], Steven Gazal[10,11], Po-Ru Loh[5], David Reich[5,8,9,12], Sriram Sankararaman[2,3,13]*

[1]Department of Computational Biology, Cornell University, New York, United States; [2]Department of Human Genetics, University of California, Los Angeles, Los Angeles, United States; [3]Department of Computer Science, University of California, Los Angeles, Los Angeles, United States; [4]Analytical and Translational Genetics Unit, Center for Genomic Medicine, Massachusetts General Hospital, Boston, United States; [5]Program in Medical and Population Genetics, Broad Institute of MIT and Harvard, Cambridge, United States; [6]Stanley Center for Psychiatric Research, Broad Institute of MIT and Harvard, Cambridge, United States; [7]Dana-Farber Cancer Institute, Harvard Medical School, Boston, United States; [8]Department of Genetics, Harvard Medical School, Boston, United States; [9]Department of Human Evolutionary Biology, Harvard University, Cambridge, United States; [10]Center for Genetic Epidemiology, Department of Public and Population Health Sciences, University of Southern California, Los Angeles, United States; [11]Division of Genetics,Department of Medicine, Brigham and Women's Hospital, Harvard Medical School, Boston, United States; [12]Howard Hughes Medical Institute, Harvard Medical School, Boston, United States; [13]Department of Computational Medicine, University of California, Los Angeles, Los Angeles, United States

*For correspondence:
sriram@cs.ucla.edu

†These authors contributed equally to this work

Competing interest: The authors declare that no competing interests exist.

**Abstract** The genetic variants introduced into the ancestors of modern humans from interbreeding with Neanderthals have been suggested to contribute an unexpected extent to complex human traits. However, testing this hypothesis has been challenging due to the idiosyncratic population genetic properties of introgressed variants. We developed rigorous methods to assess the contribution of introgressed Neanderthal variants to heritable trait variation and applied these methods to analyze 235,592 introgressed Neanderthal variants and 96 distinct phenotypes measured in about 300,000 unrelated white British individuals in the UK Biobank. Introgressed Neanderthal variants make a significant contribution to trait variation (explaining 0.12% of trait variation on average). However, the contribution of introgressed variants tends to be significantly depleted relative to modern human variants matched for allele frequency and linkage disequilibrium (about 59% depletion on average), consistent with purifying selection on introgressed variants. Different from previous studies (McArthur et al., 2021), we find no evidence for elevated heritability across the phenotypes examined. We identified 348 independent significant associations of introgressed Neanderthal variants with 64 phenotypes. Previous work (Skov et al., 2020) has suggested that a majority of such associations are likely driven by statistical association with nearby modern human variants that are the true causal variants. Applying a customized fine-mapping led us to identify 112 regions across 47 phenotypes containing 4303 unique genetic variants where introgressed variants are highly likely to have a phenotypic effect. Examination of these variants reveals their substantial impact on genes that are important for the immune system, development, and metabolism.

## Editor's evaluation

Humans whose genetic ancestors lived outside Africa have a small proportion of the genome that traces back to interbreeding events with Neanderthals. To quantify the contribution of this ancestry to present-day phenotypic variation, the authors develop a convincing set of approaches that takes into account various complicating factors and apply it to a subset of the UK Biobank individuals. The work is an important contribution to human evolution and evolutionary biology more generally.

## Introduction

Genomic analyses have revealed that present-day non-African human populations inherit 1–4% of their genetic ancestry from introgression with Neanderthals (*Green et al., 2010*; *Prüfer et al., 2014*). This introgression event introduced uniquely Neanderthal variants into the ancestral out-of-Africa human gene pool, which may have helped this bottleneck population survive the new environments they encountered (*Mendez et al., 2012*; *Abi-Rached et al., 2011*; *Sankararaman et al., 2014*; *Vernot and Akey, 2014*; *Racimo et al., 2015*; *Gittelman et al., 2016*). On the other hand, many Neanderthal variants appear to have been deleterious in the modern human genetic background leading to a reduction in Neanderthal ancestry in conserved genomic regions (*Sankararaman et al., 2014*; *Vernot and Akey, 2014*; *Harris and Nielsen, 2016*; *Juric et al., 2016*; *Petr et al., 2019*). Systematically studying these variants can provide insights into the biological differences between Neanderthals and modern humans and the evolution of human phenotypes in the 50,000 y since introgression.

In principle, studying Neanderthal-derived variants in large cohorts of individuals measured for diverse phenotypes can help understand the biological impact of Neanderthal introgression. Previously, (*Dannemann and Kelso, 2017*) showed that some Neanderthal introgressed variants are significantly associated with traits such as skin tone, hair color, and height based on genome-wide association studies (GWAS) in British samples. However, using data from Iceland, *Skov et al., 2020* found that most of the significantly associated Neanderthal introgressed single-nucleotide polymorphisms (SNPs) are in the proximity of strongly associated non-archaic variants. They suggested that these associations at Neanderthal introgressed SNPs were driven by the associations at linked non-archaic variants, indicating a limited contribution to modern human phenotypes from Neanderthal introgression. In contrast to these attempts to associate individual introgressed variants with a trait, studies have attempted to measure the aggregate contribution of introgressed Neanderthal SNPs to trait variation (*Simonti et al., 2016*; *McArthur et al., 2021*). A recent study by *McArthur et al., 2021* estimated the proportion of heritable variation that can be attributed to introgressed variants though their approach is restricted to common variants (minor allele frequency >5%) that represent a minority of introgressed variants. Despite these attempts, assessing the contribution of introgressed Neanderthal variants towards specific phenotypes remains challenging. The first challenge is that variants introgressed from Neanderthals are rare on average (due to the low proportion of Neanderthal ancestry in present-day genomes). The second challenge arises from the unique evolutionary history of introgressed Neanderthal variants, resulting in distinct population genetic properties at these variants, which can, in turn, confound attempts to characterize their effects. As a result, attempts to characterize the systematic impact of introgressed variants on complex phenotypes need to be rigorously assessed.

To enable analyses of genome-wide introgressed Neanderthal variants in large sample sizes, we selected and added SNPs that tag introgressed Neanderthal variants to the UK Biobank Axiom Array that was used to genotype the great majority of the approximately 500,000 individuals in the UK Biobank (UKBB) (*Bycroft et al., 2018*). We used a previously compiled map of Neanderthal haplotypes in the 1000 Genomes European populations (*Sankararaman et al., 2014*) to identify introgressed SNPs that tag these haplotypes. After removing SNPs that are well-tagged by those previously present on the UKBB array, we used a greedy algorithm to select 6027 SNPs that tag the remaining set of introgressed SNPs at $r^2 > 0.8$, which were then added to the UKBB genotyping array to better tag Neanderthal ancestry. These SNPs allow variants of Neanderthal ancestry to be confidently imputed and allow us to identify a list of 235,592 variants that are likely to be Neanderthal-derived (termed Neanderthal Informative Mutations [NIMs]) out of a total of 7,774,235 QC-ed SNPs in UKBB (see 'Methods'; Appendix 1).

The goals of our study are threefold: (1) to estimate the contribution of NIMs to phenotypic variation in modern humans, (2) to test the null hypothesis that an NIM has the same contribution to phenotypic variation as a non-introgressed modern human SNP, and (3) to pinpoint regions of the genome at which NIMs are highly likely to modulate phenotypic variation. We develop rigorous methodology for each of these goals that we validate in simulations. We then applied these methods to 96 distinct phenotypes measured in about 300,000 unrelated white British individuals in UKBB.

## Results

### The contribution of Neanderthal introgressed variants to trait heritability

To understand the contribution of Neanderthal introgressed variants to trait variation, we aim to estimate the proportion of phenotypic variance attributed to NIMs (NIM heritability) and to test the null hypothesis that per-NIM heritability is the same as the heritability of a non-introgressed modern human (MH) SNP. We first annotated each of the 7,774,235 QC-ed SNPs in UKBB as either an NIM or an MH SNP (see 'Methods'). NIMs include SNPs created by mutations that likely originated in the Neanderthal lineage after the human-Neanderthal split. SNPs that are not defined as NIMs are annotated as MH SNPs that likely originated in the modern human lineage or the human-Neanderthal common ancestor.

To estimate NIM heritability, we used a recently proposed method (RHE-mc) that can partition the heritability of a phenotype measured in large samples across various genomic annotations

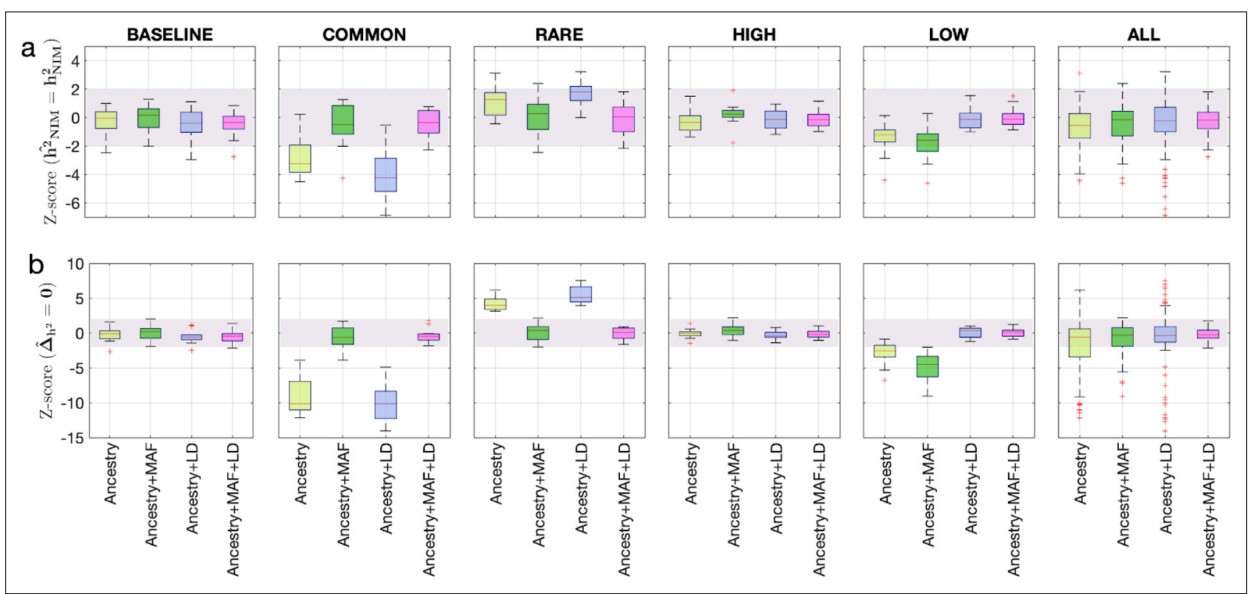

**Figure 1.** Benchmarking approaches for estimating the heritability components of Neanderthal introgression. We group simulations by relationships between minor allele frequency (MAF) and local linkage disequilibrium at an SNP on effect size (MAF-LD coupling): BASELINE, COMMON, RARE, HIGH, LOW. In each group, we perform 12 simulations with varying polygenicity and heritability (see 'Methods'). Additionally, we combine results from all simulations together as ALL. We plot the distributions of two Z-scores (y-axis), one on each row: (**a**) Z-score ($\hat{h}^2_{NIM} = h^2_{NIM}$) tests whether the estimated and true Neanderthal Informative Mutations (NIM) heritability are equal, and (**b**) Z-score ($\hat{\Delta}_{h^2} = 0$) tests whether the estimated per-NIM heritability is the same as the per-SNP heritability of modern human (MH) SNPs (see 'Methods'). In each panel, we present results from a variance components analysis method (RHE-mc) using four different input annotations: ancestry only where ancestry is either NIM or MH, ancestry + MAF, ancestry + LD, ancestry + MAF + LD. A calibrated method is expected to have Z-scores distributed around zero and within ±2 (shaded region). Among all tested approaches, only RHE-mc with ancestry + MAF + LD annotations is calibrated across simulations.

The online version of this article includes the following source data and figure supplement(s) for figure 1:

**Source data 1.** RHE-mc results in simulated data.

**Figure supplement 1.** Benchmarking different methods for estimating the total SNP heritability.

**Figure supplement 2.** Estimating the heritability components of Neanderthal introgression under a genetic architecture in which rare variants are enriched for phenotypic effects.

(*Pazokitoroudi et al., 2020*). We applied RHE-mc with genomic annotations that correspond to the ancestry of each SNP (NIM vs. MH) to estimate NIM heritability ($h^2_{NIM}$). We also attempted to estimate whether per-NIM heritability is the same as the per-SNP heritability of MH SNPs ($\Delta_{h^2}$). A positive (negative) value of $\Delta_{h^2}$ indicates that, on average, an NIM makes a larger (smaller) contribution to phenotypic variation relative to a MH SNP.

To assess the accuracy of this approach, we performed simulations where NIMs are neither enriched nor depleted in heritability (true $\Delta_{h^2} = 0$). Following previous studies of the genetic architecture of complex traits (*Evans et al., 2018*; *Evans et al., 2018*), we simulated phenotypes (across 291,273 unrelated white British individuals and 7,774,235 SNPs) with different architectures where we varied heritability, polygenicity, and how the effect size at a SNP is coupled to its population genetic properties (the minor allele frequency [MAF] at the SNP and the linkage disequilibrium or LD around an SNP). We explored different forms of MAF-LD coupling where BASELINE assumes that SNPs with phenotypic effects are chosen randomly, RARE (COMMON) assumes that rare (common) variants are enriched for phenotypic effects, and HIGH (LOW) assumes that SNPs with high (low) levels of LD (as measured by the LD score; *Finucane et al., 2015*) are enriched for phenotypic effects (see 'Methods'). Estimates of $h^2_{NIM}$ and $\Delta_{h^2}$ tend to be miscalibrated (*Figure 1ab*). The miscalibration is particularly severe when testing $\Delta_{h^2}$ so that a test of the null hypothesis has a false-positive rate of 0.55 across all simulations (at a p-value threshold of 0.05).

To understand these observations, we compared the MAFs and LD scores at NIMs to MH SNPs. We observe that NIMs tend to have lower MAF (*Figure 2a*) and higher LD scores compared to MH SNPs (*Figure 2b*) (the average MAF of NIMs and MH SNPs are 3.9% and 9.9%, respectively, while their average LD scores are 170.6 and 64.9). Among the QC-ed SNPs, 76.9% of NIMs have MAF >1%, and 27.7% have MAF >5%, in contrast to 61.6% and 41.6% of MH SNPs. Distinct from MH SNPs, the MAF and LD score of NIMs tend not to increase with each other (*Figure 2cd*). We replicated this observation using NIMs that had been identified by an alternate approach (*McArthur et al., 2021*; Appendix 5).

To account for the differences in the MAF and LD scores across NIMs and MH SNPs, we applied RHE-mc with annotations corresponding to the MAF and the LD score at each SNP (in addition to the ancestry annotation that classifies SNPs as NIM vs. MH) to estimate NIM heritability ($h^2_{NIM}$) and to test whether per-NIM heritability is the same as the per-SNP heritability of MH SNPs, that is, $\Delta_{h^2} = 0$ (see 'Methods,' 'Appendix 4'). Our simulations show that RHE-mc with SNPs assigned to annotations that account for both MAF and LD (in addition to the ancestry annotation that classifies SNPs as NIM vs. MH) is accurate both in the estimates of $h^2_{NIM}$ (*Figure 1a*) and in testing the null hypothesis that $\Delta_{h^2} = 0$ (the false positive rate of a test of $\Delta_{h^2} = 0$ is 0.017 at a p-value threshold of 0.05; *Figure 1b*). On the other hand, not accounting for either MAF or LD leads to poor calibration (*Figure 1*; we observe qualitatively similar results when estimating genome-wide SNP heritability; *Figure 1—figure supplement 1*). To further assess the robustness of our results to the genetic architecture, we also performed simulations under a model that assumes an even greater enrichment of SNP effects among rare variants wherein SNPs with MAF <1% constitute 90% of the causal variants (ULTRA RARE). RHE-mc with MAF and LD annotations remains accurate in its estimates of $h^2_{NIM}$ and in testing the null hypothesis that $\Delta_{h^2} = 0$ (*Figure 1—figure supplement 2*).

We then applied RHE-mc with ancestry + MAF + LD annotations to analyze a total of 96 UKBB phenotypes that span 14 broad categories. In all our analyses, we include the top 5 PCs estimated from NIMs (NIM PCs) as covariates in addition to the top 20 genetic PCs estimated from common SNPs, sex, and age (see 'Methods'). The inclusion of NIM PCs is intended to account for stratification at NIMs that may not be adequately corrected by including genotypic PCs estimated from common SNPs (we also report concordant results from our analyses when excluding NIM PCs; 'Appendix 3' and *Figure 4—figure supplement 1*).

We first examined NIM heritability to find six phenotypes with significant NIM heritability (Z-score $\left( \widehat{h^2}_{NIM} = 0 \right) > 3$): body fat percentage, trunk fat percentage, whole body fat mass, overall health rating, gamma glutamyltransferase (a measure of liver function), and forced vital capacity (FVC) (*Figure 3a and c*). Meta-analyzing within nine categories that contain at least four phenotypes, we find that $meta - \widehat{h^2}_{NIM}$ is significantly larger than zero for anthropometry, blood biochemistry, bone densitometry, kidney, liver, and lung but not for blood pressure, eye, lipid metabolism (p<0.05 accounting for the number of hypotheses tested). Meta-analyzing across all phenotypes with low correlation, we

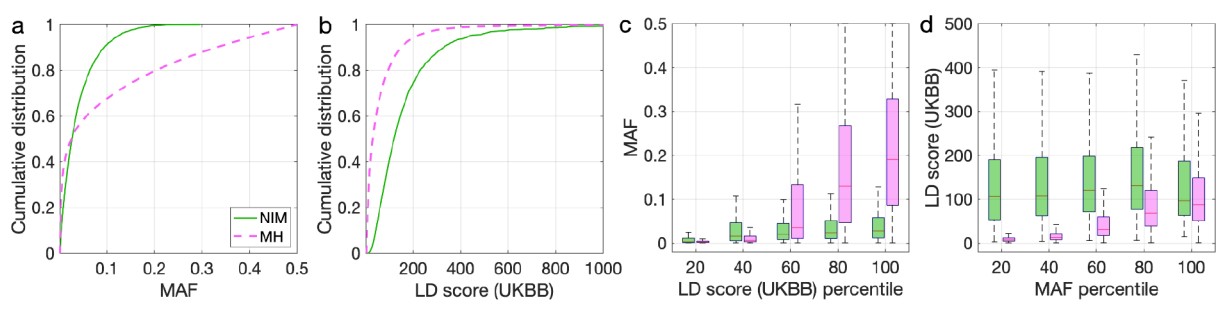

**Figure 2.** Distributions of minor allele frequency (MAF) and LD-score in Neanderthal Informative Mutations (NIMs) and modern human (MH) SNPs. Empirical cumulative distribution functions of (**a**) MAF and (**b**) LD scores of NIMs (in solid green line) and MH SNPs (in pink dashed line) estimated in the UK Biobank (UKBB). (**c**) Boxplots of MAFs of NIMs (on the left filled in green) and MH SNPs (on the right side filled in pink) while controlling for LD score (UKBB). (**d**) Boxplots of LD score (UKBB) of NIMs and MH SNPs while controlling for MAF. NIMs and MH SNPs are divided by the 20, 40, 60, 80, 100 (**c**) LD score (UKBB) percentile or MAF percentile (**d**) based on all QC-ed SNPs (7,774,235 imputed SNPs with MAF >0.001). The lower and upper edges of a box represent the first and third quartile (qu1 and qu3), respectively; the horizontal red line inside the box indicates median (md); the whiskers extend to the most extreme values inside inner fences, md ± 1.5 (qu3–qu1).

obtain overall NIM heritability estimates ($meta - \widehat{h^2}_{NIM}$) = 0.12% (one-sided p=6.6×10⁻³¹). The estimates of NIM heritability are modest as would be expected from traits that are highly polygenic and given that NIMs account for a small percentage of all SNPs in the genome (see 'Methods).

We next tested whether the average heritability at an NIM is larger or smaller compared to a MH SNP ($\hat{\Delta}_{h^2} = 0$). We find 17 phenotypes with significant evidence of depleted NIM heritability that include standing height, body mass index, and HDL cholesterol (Z-score < –3; **Figure 3b and d**). Five phenotypic categories show significant NIM heritability depletion (anthropometry, blood biochemistry, blood pressure, lipid metabolism, lung) in meta-analysis. Meta-analyzing across phenotypes, we find a significant depletion in NIM heritability ($meta - \hat{\Delta}_{h^2}$ = –1.7 × 10⁻³, p=2.1×10⁻³⁶). On average, we find that heritability at NIMs is reduced by about 57% relative to an MH variant with matched MAF and LD characteristics. In contrast to the evidence for depletion in NIM heritability, we find no evidence for traits with elevated NIM heritability across the phenotypes analyzed. We repeated these analyses using NIMs that had been identified using a different approach (**Browning et al., 2018**) and obtained concordant results ('Appendix 5'). Our observations are consistent with NIMs having been primarily under purifying selection for thousands of generations (**Harris and Nielsen, 2016**; **Petr et al., 2019**). Nevertheless, as evidenced by their overall heritability, NIMs still make a significant contribution to phenotypic variation in present-day humans.

We also investigated the impact of controlling for MAF and LD on our findings in UKBB. Analyses that do not control for MAF and LD tend to broadly correlate with our results that control for both (Pearson's r = 0.96, 0.68, and 0.65 and p<10⁻¹² among $\widehat{h^2}$, $\widehat{h^2}_{NIM}$, and $\hat{\Delta}_{h^2}$). However, these analyses underestimate both heritability (**Figure 4a**) and NIM heritability (**Figure 4b**), resulting in apparent NIM heritability depletion (Z-score < –3) in 83 of the 96 phenotypes (**Figure 4c**). While yielding qualitatively similar conclusions about the depletion in heritability at NIMs relative to MH SNPs, prior knowledge that per SNP heritability of complex traits can be MAF and LD dependent (**Evans et al., 2018**) coupled with our extensive simulations lead us to conclude that controlling for MAF and LD leads to more accurate results.

An interesting hypothesis is whether the depletion in heritability that we observe here reflects selection specifically against Neanderthal alleles or whether these could represent selection against functional changes in general since prior work has shown that Neanderthal alleles tend to be distributed further away from regions of the genome under selective constraint (**Sankararaman et al., 2014**; **Juric et al., 2016**). To answer this question, we can compare the average heritability at NIMs to modern human SNPs matched for B-value, a measure of background selection (**McVicker et al., 2009**). We attempted to estimate the difference in average heritability between NIMs and MH SNPs ($\Delta_{h^2}$) while matching on quartiles of B-value bins, in addition to MAF and LD bins. A challenge with this approach is the large number of annotations leads to annotations with few SNPs so that $h^2_{NIM}$ estimates are substantially less precise (estimated with standard errors that are about 10 times larger on average

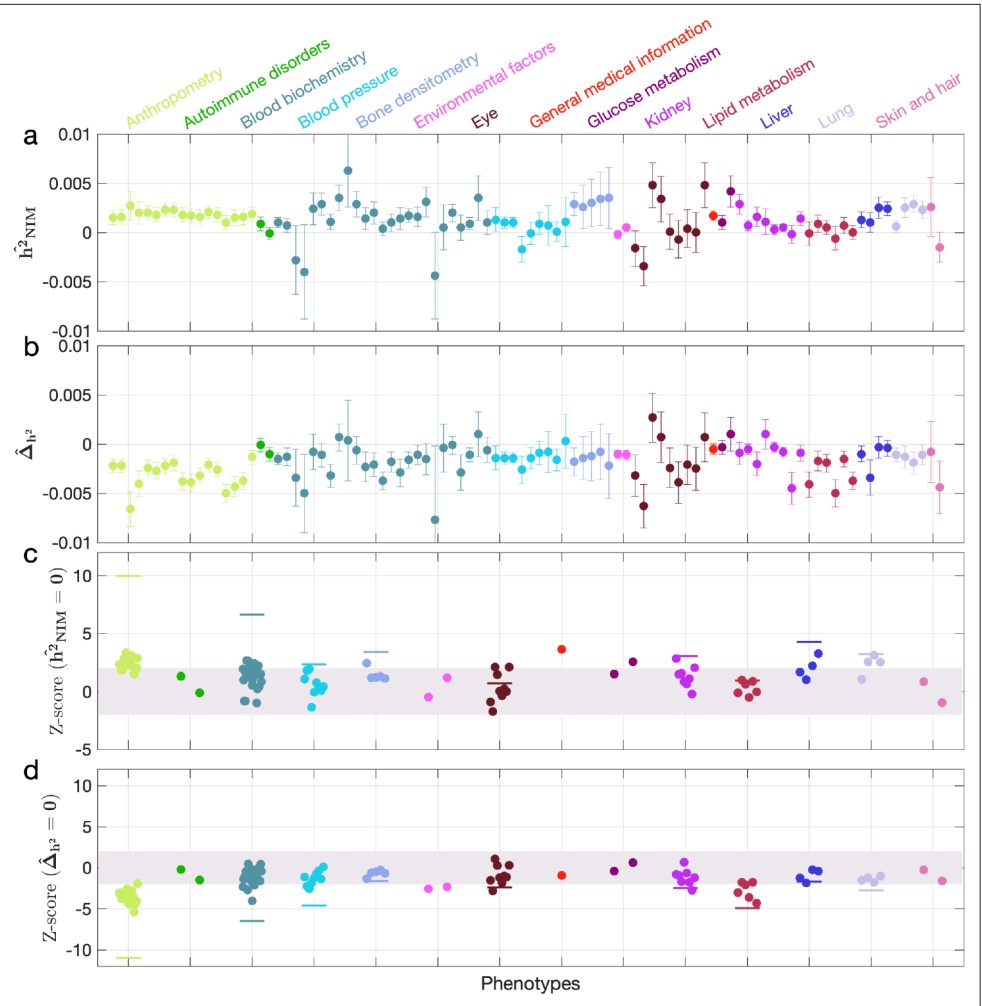

**Figure 3.** Neanderthal Informative Mutation (NIM) heritability in UK Biobank (UKBB) phenotypes. (**a**) Estimates of NIM heritability ($\widehat{h^2}_{NIM}$) and (**c**) the Z-score of $\widehat{h^2}_{NIM}$ (testing the hypothesis that NIM heritability is positive) for each UKBB phenotype. Analogously, (**b**) estimates of $\hat{\Delta}_{h^2}$ and Z-score (**d**) of $\hat{\Delta}_{h^2}$ (testing the hypothesis that per-NIM heritability is equal to per-SNP heritability at modern human [MH] SNPs after controlling for MAF and LD). Phenotypic categories are shown in alphabetical order and listed on the top of panel (**a**) in the same color and alphabetical order (from top to bottom, and left to right) as they are in the figure. The estimate for each phenotype is shown as one colored dot, on the x-axis based on its phenotypic category, and on the y-axes based on its Z-score ($\widehat{h^2}_{NIM} = 0$) and Z-score ($\hat{\Delta}_{h^2} = 0$), for panels (**c**) and (**d**) respectively. For each phenotypic category with at least four phenotypes, their Z-scores from random effect meta-analysis are plotted with the flat colored lines (see 'Methods'). The color shades cover Z-scores around zero and within ±2. g.

The online version of this article includes the following source data and figure supplement(s) for figure 3:

**Source data 1.** UKBB phenotype annotation.

**Source data 2.** RHE-mc results with Ancestry+MAF+LD annotations and NIM PCs included in covariates applied to 96 UKBB phenotypes.

**Figure supplement 1.** Neanderthal Informative Mutation (NIM) heritability in UK Biobank (UKBB) phenotypes after accounting for background selection.

---

than in the setting where we do not match on B-values). Consequently, we do not find a significant difference in the per-SNP heritability at NIMs compared to MH SNPs. Instead, we estimated ($\Delta_{h^2}$) matching on B-values and MAF having confirmed that the $h^2_{NIM}$ estimates are estimated with precision comparable to the setting where we do not match on B-values. In this setting, we continue to observe a significant depletion in NIM heritability across phenotypes (55 phenotypes with Z-score < –3) with no evidence for traits with elevated (*Figure 3—figure supplement 1*).

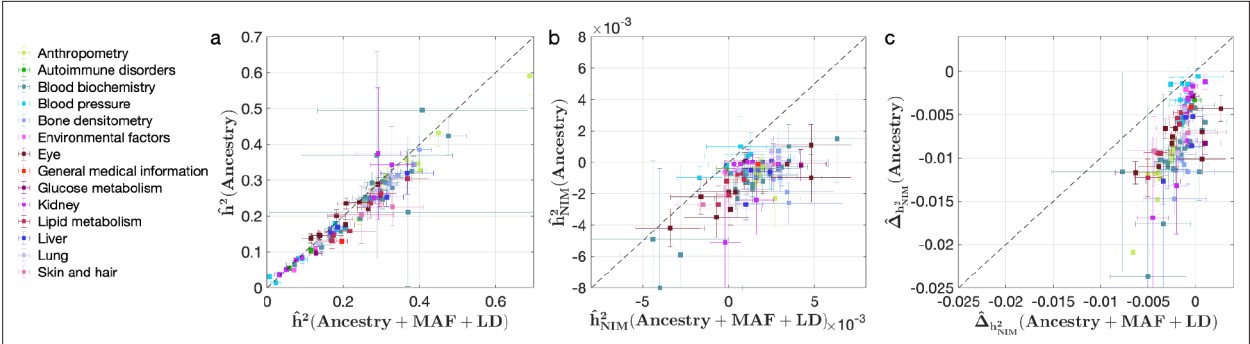

**Figure 4.** Comparing heritability analyses with and without controlling for minor allele frequency (MAF) and LD in UK Biobank (UKBB) phenotypes. Each phenotype is shown with one dot colored by the phenotypic category it belongs to, on the y-axis based on its point estimate and standard error (estimated by RHE-mc with Ancestry annotation) and on the x-axis based on its point estimate and standard error (estimated by RHE-mc with ancestry + MAF + LD annotation). Estimates shown are (**a**) total heritability $\widehat{h^2}$, (**b**) Neanderthal Informative Mutation (NIM) heritability $\widehat{h^2}_{NIM}$, and (**c**) the difference between per-NIM heritability and matched modern human (MH) SNPs heritability $\hat{\Delta}_{h^2}$. Not controlling for MAF and LD leads to underestimation of NIM heritability, which leads to false positives when testing whether heritability at an NIM is elevated or depleted relative to an MH SNP.

The online version of this article includes the following source data and figure supplement(s) for figure 4:

**Source data 1.** RHE-mc results with Ancestry only annotation and NIM PCs included in covariates applied to 96 UKBB phenotypes.

**Figure supplement 1.** Comparing heritability estimates from RHE-mc without controlling for Neanderthal Informative Mutation (NIM) principal components (PCs) with Ancestry + MAF + LD annotation and RHE-mc with Ancestry annotation in UK Biobank (UKBB) phenotypes.

Taken together, our analyses suggest that depletion in heritability likely reflects selection against Neanderthal alleles rather than selection against variation in functionally constrained regions of the genome in general.

## Identifying genomic regions at which introgressed variants influence phenotypes

Having documented an overall contribution of NIMs to phenotypic variation, we focus on identifying individual introgressed variants that modulate variation in complex traits. We first tested individual NIMs for association with each of 96 phenotypes (controlling for age, sex), 20 genetic PCs (estimated from common SNPs), and 5 NIM PCs (that account for potential stratification that is unique to NIMs). We obtained a total of 13,075 significant NIM-phenotype associations in 64 phenotypes with 8018 unique NIMs (p<10⁻¹⁰ that accounts for the number of SNPs and phenotypes tested) from which we obtain 348 significant NIM-phenotype associations with 294 unique NIMs after clumping associated NIMs by LD (see 'Methods).

A limitation of the association testing approach is the possibility that an NIM might appear to be associated with a phenotype simply due to being in LD with a non-introgressed variant (*Skov et al., 2020*). We formally assessed this approach in simulations of phenotypes with diverse genetic architectures described previously where the identities of causal SNPs are known. An NIM that was found to be associated with a phenotype (p<10⁻¹⁰) was declared a true positive if the 200 kb region surrounding the associated NIM contains any NIM with a non-zero effect on the phenotype and a false positive otherwise. Averaging across all genetic architectures, the false discovery proportion (FDP; the fraction of false positives among the significant NIMs) of the association testing approach is around 30% (*Figure 5b*). Hence, finding NIMs that are significantly associated with a phenotype does not confidently localize regions at which introgressed variants affect phenotypes.

To improve our ability to identify NIMs that truly modulate phenotype, we designed a customized pipeline that combines association testing with a fine-mapping approach that integrates over the uncertainty in the identities of causal SNPs to identify sets of NIMs that plausibly explain the association signals at a region (*Figure 5a*). Our pipeline starts with a subset of significantly associated NIMs that are relatively independent (p<10⁻¹⁰) followed by the application of a statistical fine-mapping method (SuSiE) within the 200 kb window around each NIM signal (*Wang et al., 2020*) and additional post-processing to obtain a set of NIMs that have an increased probability of being causal for a trait.

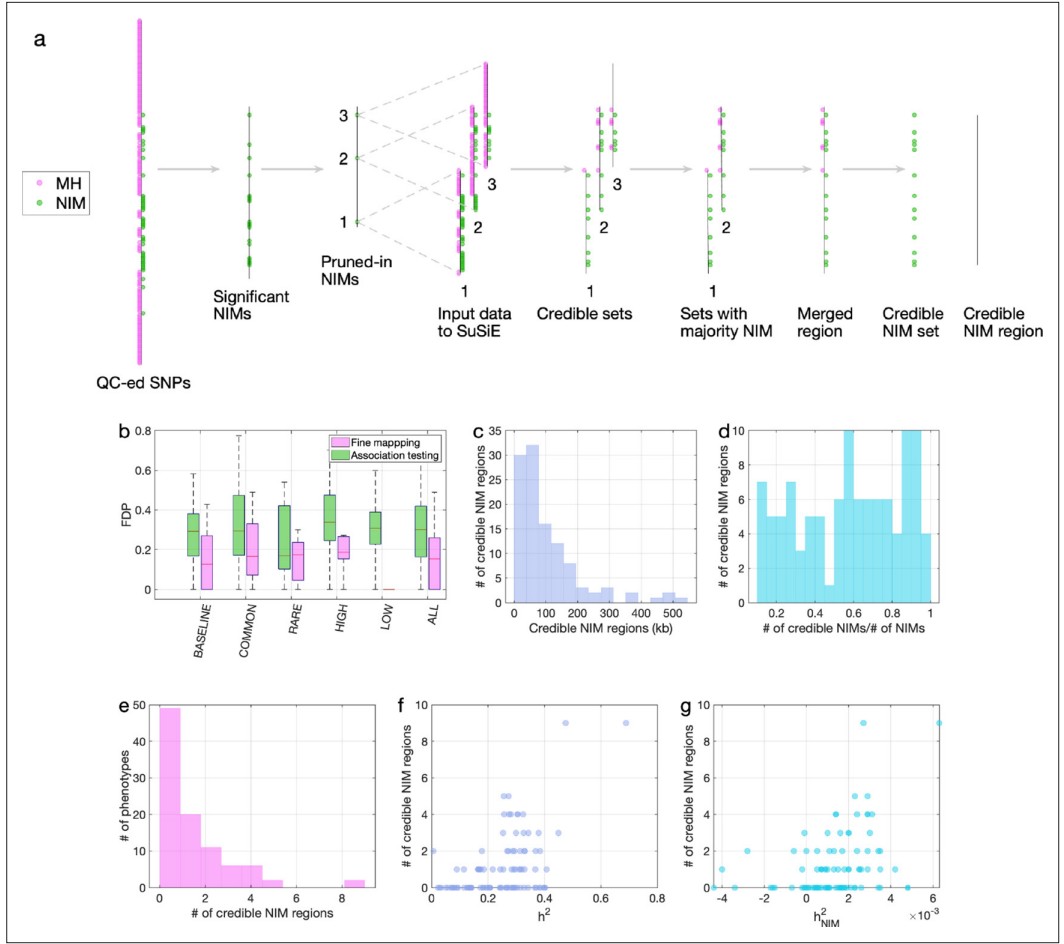

**Figure 5.** Fine mapping of Neanderthal Informative Mutations (NIMs) in simulations and the UK Biobank (UKBB). (**a**) Fine mapping pipeline to identify NIMs that aims to identify genomic regions at which NIMs are likely to modulate phenotypic variation (credible NIM regions). (**b**) Comparison of approaches for identifying credible NIM regions. For each simulation, false discovery proportion (FDP) is computed for association testing compared to our pipeline (combining association testing and fine-mapping). The distributions of the FDP are shown across genetic architectures (summarized across groupings of coupling of effect size, minor allele frequency [MAF] and LD) and summarized across architectures (ALL). Our approach to identifying credible NIMs decreases FDP in all studied architectures (the LOW LD setting has a median and quartiles of zero across replicates). (**c**) The distribution of the length of credible NIM regions across 96 UKBB phenotypes. (**d**) Distribution of the ratio between the number of credible NIMs and number of tested NIMs (in the example of panel (**a**), the number of tested NIMs is the union of NIMs in input to the fine-mapping software (SuSiE) 1 and 2) showing that our approach is effective in prioritizing NIMs that affect phenotype. (**e**) The distribution of the number of credible NIM regions among phenotypes. The number of credible NIM regions is positively correlated with (**f**) heritability (**g**) NIM heritability.

The online version of this article includes the following source data for figure 5:

**Source data 1.** Fine mapping FDP in simulated data.

We term the NIMs within this set credible NIMs while the shortest region that contains all credible NIMs in a credible set is termed the credible NIM region (see 'Methods; *Figure 5a*).

We employed the same simulations as previously described to evaluate our fine-mapping approach. The fine mapping approach yields a reduction in the FDP relative to association mapping (FDP of 15.6% on average; *Figure 5b*) while attributing the causal effect to a few dozen NIMs within the credible NIM set (mean: 79, median: 54 NIMs across all simulations). Applying our pipeline to the set of 96 UKBB phenotypes, we identified a total of 112 credible NIM regions containing 4303 unique credible NIMs across 47 phenotypes (*Figure 6a*). The median length of credible NIM regions, 65.7 kb (95% CI: [4.41 kb, 469.3 kb]) is close to the expected length of Neanderthal intro-gressed segments (*Skov et al., 2020*) suggesting that the resolution of our approach is that of an

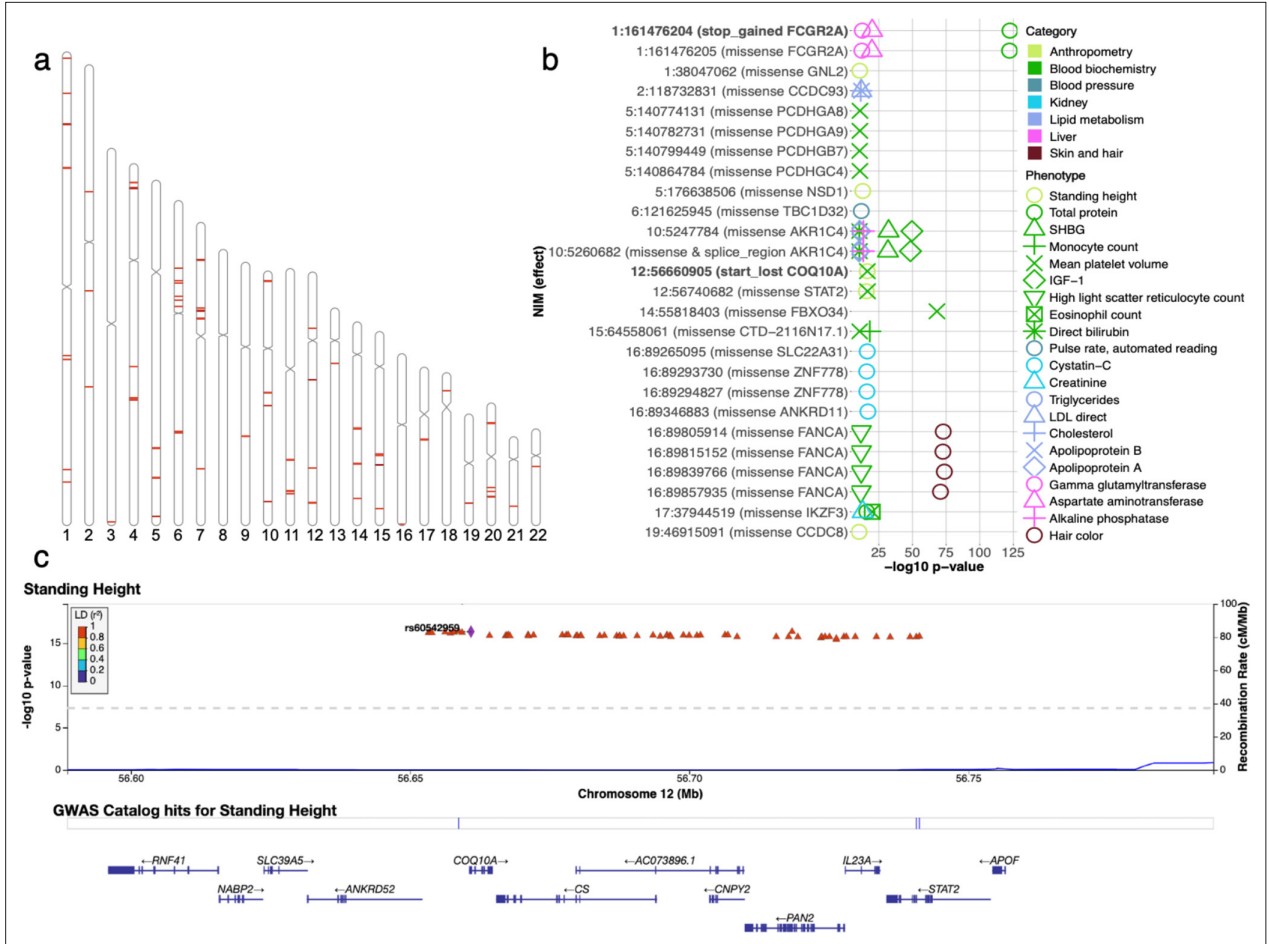

**Figure 6.** Analysis of credible Neanderthal Informative Mutations (NIMs). (**a**) Distribution of credible NIMs across the genome. (**b**) High and moderate impact credible NIMs annotated by SnpEff software (*Cingolani et al., 2012*). A total of 26 credible NIMs have high (marked in bold) or moderate impact effects on nearby genes (chromosome number and hg19 coordinates). The effects of the SNP and the gene name are displayed. This plot shows significant associations of these NIMs with specific phenotypes (color denotes the phenotype category). (**c**) Plot of 300 kb region surrounding rs60542959 (marked in black diamond; hg19 coordinates), a credible NIM for standing height that results in loss of the start codon in COQ10A. The plot displays other significantly associated NIMs in the region along with their LD ($r^2$) to rs60542969 in 1000 Genomes Europeans (*Boughton et al., 2021*).

The online version of this article includes the following source data and figure supplement(s) for figure 6:

**Source data 1.** 112 credible NIM sets and credible NIMs.

**Source data 2.** SnpEff annotation of all unique credible NIMs.

**Figure supplement 1.** Credible Neanderthal Informative Mutation (NIM) in the FCGR2A gene associated with gamma-glutamyl transferase levels.

**Figure supplement 2.** Credible Neanderthal Informative Mutation (NIM) in the AKR1C4 gene is associated with bilirubin levels.

**Figure supplement 3.** Number of unique credible Neanderthal Informative Mutations (NIMs) that are expression quantitative trait loci (eQTLs).

introgressed LD block (*Figure 5c*). While fine mapping generally attributes the causal signal to a subset of the tested NIMs (mean: 55.8, median: 37 NIMs across phenotypes), the degree of this reduction varies across regions likely reflecting differences in the LD among NIMs (*Figure 5d*). We do not detect any credible NIM in 49 out of 96 phenotypes potentially due to the limited power of our procedure that aims to control the FDR (*Figure 5e*). The sensitivity of our method is affected by both total heritability (*Figure 5f*, Pearson's $r = 0.49$, p=3.3×10⁻⁷) and NIM heritability (*Figure 5g*, Pearson's $r = 0.36$, p=3.3×10⁻⁴). A linear model that uses both total heritability and NIM heritability to predict the number of credible sets yields $r^2 = 0.29$, p=1.3×10⁻⁵ and 0.015, respectively, while linear models with only total heritability or only NIM heritability result in statistically lower $r^2$ (0.24 and 0.13, respectively).

## Examination of the functional impact of credible NIMs

We annotated all 4303 unique credible NIMs using SnpEff (*Cingolani et al., 2012*) to identify a total of 26 NIMs with high (e.g., start codon loss, stop codon gain) or moderate impact (nonsynonymous variants) on genes (*Figure 6b*). We identified two credible NIMs, rs9427397 (1:161,476,204 C>T) and rs60542959 (12:56,660,905 G>T), that have a high impact on protein sequences. The 1:161,476,204 C>T mutation, an NIM that is associated with increased gamma glutamyltransferase and aspartate amino-transferase (enzymes associated with liver function) and decreased total protein levels in blood, intro-duces a premature stop codon in the FCGR2A gene (*Figure 6—figure supplement 1*). FCGR2A codes for a receptor in many immune cells, such as macrophages and neutrophils, and is involved in the process of phagocytosis and clearing of immune complexes. This NIM is in a region that contains SNPs shown in several GWAS linked to rheumatoid arthritis (*Okada et al., 2014*; *Laufer et al., 2019*). The other high-impact mutation, 12:56,660,905 G>T (rs60542959), results in the loss of the start codon in COQ10A, and this SNP is a credible NIM for both mean platelet volume and standing height (*Figure 6c*). COQ10 genes (A and B) are important in respiratory chain reactions. Deficiencies of CoQ10 (MIM 607426) have been associated with encephalomyopathy, infantile multisystemic disease; cerebellar ataxia, and pure myopathy (*Quinzii et al., 2008*). The start codon in COQ10A is conserved among mammals with its loss having a potentially significant effect on COQ10A expression in immune cells (*Kubota and Suyama, 2020*).

In addition, we detect 24 credible NIMs that function as missense mutations in 19 genes. Seven out of the 19 genes are known to have immune related functions (FCGR2A, PCDHG (A8, A9, B7, C4), STAT2, and IKZF3). The NIM in STAT2 (rs2066807, 12:56,740,682 C>G) was the first adaptive introgression locus to be identified (*Mendez et al., 2012*). The STAT2 introgressed variant segregates at 0.066 frequency in the UKBB white British and leads to an I594M amino acid change in the corre-sponding protein. STAT2 gene and COQ10A are neighboring genes, thereby providing an example of an introgressed region that potentially impacts function at multiple genes (*Figure 6c*).

At least 7 of the 12 genes not known to be immune related have other important functions docu-mented in the literature, such as DNA replication/damage (FANCA, CCDC8) (*Moldovan and D'An-drea, 2009*; *Jiang et al., 2016*), transition in meiosis (FBXO34) (*Zhao et al., 2021*), detoxification/ metabolism (AKR1C4) (*Lee et al., 2009*), and neurological/developmental (ZNF778, ANKRD11, TBC1D32) functions (*Willemsen et al., 2010*; *Alves et al., 2019*; *Hietamäki et al., 2020*). rs17134592 (10:5260682 C>G) is a non-synonymous mutation in AKR1C4, a gene that is involved in the metab-olism of ketone-containing steroids in the liver. The NIM is associated with increased serum bilirubin levels (p=3×10⁻¹¹) (*Figure 6—figure supplement 2a*) while also being associated with increased levels of alkaline phosphatase, insulin-like growth factor 1 (IGF1) and decreased apolipoprotein A, sex hormone binding globulin (SHBG) and triglyceride levels. rs17134592 has been identified to be a splicing QTL that is active in the liver and testis in the GTeX data (*Figure 6—figure supplement 2b*). This NIM alters leucine to valine (L311V), which, in combination with the tightly linked non-synonymous variant rs3829125 (S145C) in the same gene, have been shown to confer a three- to fivefold reduction in catalytic activity of the corresponding enzyme (3-alpha hydroxysteroid dehydrogenase) in human liver (*Kume et al., 1999*). Interestingly, the single amino acid change S145C did not significantly alter enzyme activity, suggesting the importance of the amino acid residue at position 311 for the substrate binding of the enzyme.

To assess the role of noncoding NIMs that impact phenotype, we investigated the prevalence of expression quantitative trait loci (eQTLs) among credible NIMs. We used FUMA (*Watanabe et al., 2017*) to annotate whether a credible NIM is an eQTL using eQTLs identified across the 54 tissues analyzed in GTEx v8. We find that 60.8% of the credible NIMs are eQTLs for at least one gene in at least one tissue while only 25.6% of all NIMs are eQTLs. Out of 112 credible NIM sets, 23 have at least one credible NIM that alters coding sequence (*Figure 6b*), 79 have at least one credible NIM that works as an eQTL in at least one tissue while 22 have at least one credible NIM that impacts both coding sequence and gene expression. Additionally, we listed the GTEx v8 tissues where credible NIMs are found to be eQTLs for genes expressed in those tissues (*Figure 6—figure supplement 3*). We find examples of credible NIMs for specific phenotypes that are eQTLs in relevant tissues: the credible set for measures of lung capacity (forced expiratory volume [FEV1] and forced vital capacity [FVC]) contains eQTLs for gene expressed in lung while credible sets for a measure of liver function (alanine aminotransferase levels in blood) contain eQTLs in liver.

## Discussion

Our analysis demonstrates the complex influence of Neanderthal introgression on complex human phenotypes. The assessment of the overall contribution of introgressed Neanderthal alleles to phenotypic variation indicates a pattern where, taken as a group, these alleles tend to be depleted in their impact on phenotypic variation (with about a third of the studied phenotypes showing evidence of depletion). This pattern is consistent with these alleles having entered the modern human population roughly 50,000 y ago and being subject to purifying selection. Selection to purify deleterious introgressed variants, coupled with stabilizing selection on human complex traits, could result in introgressed heritability depletion such that the remaining introgressed variants in present-day humans tend to have smaller phenotypic effects compared to other modern human variants.

In contrast to the previous heritability analyses by *McArthur et al., 2021*, we did not find any NIM heritability enrichment in the 96 phenotypes. This discrepancy could be due to the different methods and NIMs used in the two studies. McArthur et al. estimate the heritability associated with common NIMs (NIMs with MAF >5%) using stratified LD score regression (S-LDSR) with LD scores computed from 1KG (see Appendix 2). Because more than 70% of NIMs have MAF <5%, this approach may not extrapolate to understand the heritability from all NIMs. An additional potential concern with analyses of NIMs is the possibility of confounding due to population structure among these introgressed variants. Typical approaches to account for population stratification based on the inclusion of principal components (PCs) may not be adequate as these PCs are computed from common SNPs on the UKBB genotyping array and may not account for stratification at the NIMs that tend to be rare on average (*Mathieson and McVean, 2012*). Since our analyses work directly on individual genotype data, we are better able to control for stratification specific to NIMs by including PCs estimated from NIMs in addition to PCs estimated from common SNPs (with the caveat that even this approach is not guaranteed to correct for some types of stratification that can impact NIMs). In spite of these differences in methods and NIMs analyzed, our observation of an overall pattern of depletion in the heritability of introgressed alleles is consistent with the findings of McArthur et al. The robustness of this pattern might provide insights into the nature of selection against introgressed alleles.

Beyond characterizing aggregate effects of NIMs, we also attempted to identify individual NIMs that modulate phenotypic variation. A challenge in identifying such variants comes from the fact that NIMs tend to have lower MAF and higher LD compared to MH SNPs. Lower MAF tends to limit the power to detect a genetic effect while higher LD makes it harder to identify the causal variant. These challenges led us to design a fine mapping strategy for prioritizing causal NIMs that enables the identification of sets of NIMs that can credibly exert influence on specific phenotypes. Using this approach, we identified credible NIMs in a number of functionally important genes, including a premature stop codon in the FCGR2A gene, and a start codon loss in COQ10A. In addition, variants in STAT2 are found to be highly pleiotropic. As many of the genes are relevant to immune, metabolic, and developmental disorders, with functions relevant to the transition to new environments, the credible NIMs reported in our study offer a starting point for detailed investigation of the biological effects of introgressed variants. Greenbaum et al. hypothesized that introgression-based transmission of alleles related to the immune system could have helped human out-of-Africa expansion in the presence of new pathogens (*Greenbaum et al., 2019*). While our results do not directly support this hypothesis, they pinpoint introgressed alleles in immune-related genes that could have and continue to modulate human phenotypes consistent with findings from prior studies (*Abi-Rached et al., 2011*; *Mendez et al., 2012*; *Quach et al., 2016*; *Nédélec et al., 2016*; *Enard and Petrov, 2018*). Although we identified a number of likely causal NIMs in fine mapping, our strategy likely only picks up a small fraction of the functional NIMs, suggesting that additional NIMs that are causal for specific traits remain to be discovered.

Our study has several limitations due to the current availability of data and statistical methods. First, all of our analyses focus on the white British individuals in the UKBB due to the large sample size that permits the interrogation of low-frequency NIMs and our choice of NIMs based on introgressed variants segregating in European populations. Whole-genome sequencing data in diverse populations can potentially elucidate the impact of Neanderthal introgression in other out-of-African populations that harbor substantial Neanderthal ancestry. Alternatively, designing arrays that have SNPs informative of archaic ancestry followed by genotype imputation could be a fruitful strategy to leverage large Biobanks to systematically explore the contribution of archaic introgression. Second, our findings of

lower per-SNP heritability at NIMs relative to MH SNPs are consistent with two non-exclusive hypotheses: that introgressed variants tend to have a lower proportion of causal SNPs or lower effect sizes compared to non-introgressed variants. While our approach, based on heritability estimation, cannot distinguish these hypotheses, recent approaches that enable estimation of polygenicity of complex traits hold promise in this regard (*Zhang et al., 2018*; *Johnson et al., 2021*; *O'Connor, 2021*). Third, while our approach to localize credible NIMs yields a list of NIMs that are highly likely to modulate variation in a trait, our method only identifies a subset of causal variants. The design of fine mapping methods to study introgressed variants while taking into account the ancestry (as well as better incorporating other measures such as posterior inclusion probabilities) is an important direction for future work. More broadly, the unique evolutionary history of introgressed variants motivate the development of methods tailored to their population genetic properties. While our results suggest potential evolutionary models that explain our observations of depleted heritability at introgressed alleles, evolutionary models that can comprehensively explain our observations are lacking. A major challenge is the large space of potential models that need to be explored. Nevertheless, proposing and validating such models will be an important direction for future work.

## Methods

### Identification and design of SNPs that tag Neanderthal ancestry on the UK Biobank Axiom array

We chose a subset of SNPs to add to the UK Biobank Axiom array that would tag introgressed Neanderthal alleles segregating in present-day European populations.

We began with a list of 95,462 SNPs that are likely to be Neanderthal-derived from *Sankararaman et al., 2014*. These SNPs were identified to tag confidently inferred Neanderthal haplotypes in the European individuals identified in the 1000 Genomes Phase 1 data (Appendix 1).

We winnowed down this list to 43,026 SNPs after removing ones already tagged at $r^2 > 0.8$ by SNPs on the UKBiLEVE array. We then designed a greedy algorithm to capture the remaining untagged SNPs that could still be accommodated on the array (we determined the number of oligonucleotide features that would be needed to genotype each SNP as well as the total number of features available on the array through discussions with UK Biobank Axiom array design team).

Specifically, we computed LD between all pairs of Neanderthal-derived SNPs and then iteratively picked SNPs with the highest score to add to the array where the score was computed as

$$Score_{SNP\ j} = \frac{\sum_{i=1}^{n} \left[\delta_{r^2 > 0.80}(i,j)\right]\left[Derived\ frequency_{SNP\ i}\right]}{Features\ required\ genotype\ SNP\ j}$$

Here, $\delta_{r^2 > 0.80}(i,j)$ is an indicator variable that is 1 if the squared correlation coefficient between SNPs $i$ and $j$ is >0.80 and zero otherwise. Thus, SNP $j$ is scored higher if it tags other untagged SNPs on the array. The other two terms upweight SNPs that tag other Neanderthal-derived SNPs with high derived allele frequency in Europeans and downweight SNPs by the number of oligonucleotide features required to genotype the SNP.

We iteratively chose SNPs until we obtained 6027 SNPs (requiring 16,674 features) that fully tagged the remaining set of Neanderthal-derived SNPs. These 6027 SNPs were then added to the UKBiobank Axiom array.

### UK Biobank (UKBB) genotype QC

We restricted all our analyses to a set of high-quality imputed SNPs (with a hard call threshold of 0.2 and an info score ≥0.8), which, among the 291,273 imputed genotypes of UKBB unrelated white British individuals, (1) have MAF higher than 0.001, (2) are under Hardy–Weinberg equilibrium (p>10⁻⁷), and (3) are confidently imputed in more than 99% of the genomes. Additionally, we excluded SNPs in the MHC region, resulting in a total of 7,774,235 SNP which we refer to as QC-ed SNPs.

### Identification of Neanderthal Informative Mutations

We intersected the 95,462 Neanderthal-derived SNPs identified in the 1000 Genomes European individuals with UKBB QC-ed SNPs, resulting in 70,374 variants that we term confident NIM. SNPs in high linkage disequilibrium (LD) with this set are likely introduced through Neanderthal introgression. We

expanded this set by including all QC-ed SNPs, which (1) have an $r^2$ of 0.99 or higher with any confident NIM, and (2) are located in the proximal neighborhood of any confident NIM (within 200 kb). We term this set of SNPs as expanded NIMs. On average, 80.58% of expanded NIMs match the corresponding Altai Neanderthal allele, in contrast to 2.18% of the remaining SNPs, suggesting that these SNPs are also highly informative about Neanderthal ancestry. This treatment expands the number of NIMs in the UKBB QC-ed SNPs from 70,374 (confident NIMs) to 235,592 (expanded NIMs). We primarily use this more inclusive set of SNPs in our analyses, and refer to them as NIMs in the main results. SNPs that were not part of the expanded NIMs are termed MH SNPs.

## Annotating QC-ed SNPs by MAF and LD

In addition to ancestry (Neanderthal vs. MH), we annotate each QC-ed SNP by its MAF and LD. We define five MAF-based annotations by dividing all QC-ed SNPs into five equal-sized bins by their MAFs. We similarly define five LD-based annotations by dividing all QC-ed SNPs into five equal-sized bins based on their LD-score computed from 291,273 imputed unrelated white British genotypes. In-sample LD-score is computed on QC-ed genotypes using GCTA (https://cnsgenomics.com/software/gcta/#Overview) with flags "--ld-score --ld-wind 10000".

After each QC-ed SNP is annotated with three properties – ancestry (NIMs vs. MH), MAF, and LD – we use them to construct three additional sets of annotations: ancestry + MAF, ancestry + LD, and ancestry + MAF + LD annotations, by intersecting MAF annotation with ancestry annotation, LD annotation with ancestry annotation, and all three annotations, respectively. For example, for ancestry + MAF annotation, we intersect the previously defined MAF annotation with the ancestry annotation and divide SNPs into 10 non-overlapping bins – from low to high MAF with Neanderthal ancestry (five bins) and from low to high MAF with modern human ancestry (five bins). Similarly, when SNPs are annotated with LD + ancestry, we have five LD bins with Neanderthal ancestry corresponding to five LD groups with MH ancestry.

Because NIMs tend to have low MAF and high LD-score (**Figure 2**), the sizes of the annotation bins are highly uneven. To enable reliable downstream heritability analyses, we remove the annotation bins in their entirety if they include fewer than 30 SNPs. Such exceptions only occur when SNPs are annotated based on all three annotations, that is, ancestry + MAF + LD.

## Whole-genome simulations

We simulated phenotypes based on QC-ed UKBB genotypes with the same sample size (291,273) and number of SNPs (7,774,235). In each simulation, either 10,000 variants (mimicking moderate polygenicity) or 100,000 (mimicking high polygenicity) are sampled from the QC-ed SNPs to have causal phenotypic effects while the rest of the variants have zero effect. Causal effects and phenotypes are simulated with GCTA assuming either a high SNP heritability of 0.5 or a moderate SNP heritability of 0.2.

With the simulated causal NIM variants, true NIM heritability $h^2_{NIM}$ can be computed as

$$h^2_{NIM} = \sum_i \beta^2_{NIM,i} / Var(y)$$

where phenotypes $y$ are simulated based on a set of standardized genotype data with a simple additive genetic model

$$y_j = \sum_i w_{ij}\beta_i + \varepsilon_j$$

and

$$w_{ij} = (x_{ij} - 2p_i) / \sqrt{2p_i(1 - p_i)}$$

with $x_{ij}$ being the number of reference alleles for the $i$th causal variant of the $j$th individual and $p_i$ being the frequency of the $i$th causal variant, $\beta_i$ is the allelic effect of the $i$th causal variant that is drawn independently from a standard normal distribution and $\varepsilon_j$ is the residual effect generated from a normal distribution with mean 0 and variance $Var\left(\sum_i w_{ij}\beta_i\right) / \left(1/h^2 - 1\right)$. We note that when the causal SNPs are selected at random, this is the GCTA model that has been used in genetic studies of complex traits (**Yang et al., 2010**).

Following previous work (**Evans et al., 2018**), we chose causal variants according to five different MAF and LD-dependent genetic architecture: (1) BASELINE: baseline architecture, where SNPs are randomly selected to be causal variants; (2) COMMON: common SNPs are enriched for phenotypic effects so that SNPs with MAF >0.05 contribute 90% of causal variants while rare SNPs contribute 10%; (3) RARE: rare variants are enriched for phenotypic effects such that SNPs with MAF $\leq 0.05$ contribute to 90% of causal variants while the rest contribute 10%; (4) LOW: low LD SNPs are enriched for phenotypic effects, realized as SNPs whose LD-score $\leq 10$ contribute 90% of causal variants, and the rest contribute 10%; and (5) HIGH: high LD SNPs are enriched for phenotypic effects, such that SNPs with LD-score >10 contribute 90% causal variants while the rest contribute 10%. We simulated three replicates for each genetic architecture with two different values of SNP heritability (0.2 and 0.5) and two different levels of polygenicity (10,000 and 100,000 causal variants).

## Estimating NIM heritability with RHE-mc

We are interested in estimating the proportion of phenotypic variance attributed to NIMs (true NIM heritability $h^2_{NIM}$) and evaluating if the heritability at an NIM (per-NIM heritability) is larger or smaller than that of a background MH SNP. To this end, we used a variance components model that partitions phenotypic variance across genomic annotations that include ancestry (NIM vs MH) as one of the input annotations.

We use RHE-mc, a method that can partition genetic variance across large sample sizes, to estimate NIM heritability (**Pazokitoroudi et al., 2020**). For each phenotype, we run RHE-mc, in turn, with four types of input annotations: ancestry alone, ancestry + MAF, ancestry + LD, and ancestry + MAF + LD as described above. The ancestry + MAF, ancestry + LD, and ancestry + MAF + LD annotations are intended to account for the differences in the MAF and LD properties of NIMs compared to MH SNPs.

To estimate NIM heritability, $\widehat{h^2}_{NIM}$, we combine the heritability of each bin corresponding to Neanderthal ancestry:

$$\widehat{h^2}_{NIM} = \sum_i \widehat{h^2}_{NIM,i}$$

and the heritability estimates for any bins with modern human ancestry are used to compute the total heritability from MH. Thus, when we estimate NIM heritability from RHE-mc run with ancestry + MAF annotations, we add the heritability estimates from five bins of low to high MAF NIMs.

To compare the average heritability at an NIM to the heritability of a background MH SNP that is chosen to match the NIM in terms of MAF and LD profiles, we compute the following statistic:

$$\widehat{\Delta}_{h^2} = \widehat{h^2}_{NIM} - \widehat{h^2}_{MH}$$

where $\widehat{h^2}_{MH} = \sum_i \frac{M_{NIM,i}}{M_{MH,i}} \widehat{h^2}_{MH,i}$ is the heritability of the background set matched for the MAF and LD profile of the set of NIMs. Here $M_{MH,i}$ denotes the number of MH SNPs in bin $i$ (defined according to MAF and/or LD of the MH SNPs) while $M_{NIM,i}$ denotes the number of NIMs in the corresponding bin. A more detailed justification of this statistic is provided in Appendix 4.

The standard errors (s.e.) of these statistics are computed using 100 jackknife blocks using an extension of RHE-mc that takes into account the covariance among different annotations. This new version of the RHE-mc is now available at https://github.com/alipazokit/RHEmc-coeff, (copy archived at swh:1:rev:b53cfba3f8f8dd160082dda642075302f64d46a0; **Pazoki, 2022**).

## NIM heritability and META-analysis using UKBB phenotypes

We applied RHE-mc to a total of 96 UKBB phenotypes. These phenotypes fall into 14 broader phenotypic categories: anthropometry, autoimmune disorders, blood biochemistry, blood pressure, bone densitometry, environmental factors, eye, general medical information, glucose metabolism, kidney, lipid metabolism, liver, lung, and skin and hair. For each phenotype, we use RHE-mc to estimate the NIM heritability $\widehat{h^2}_{NIM}$ and the difference between per-NIM heritability and the per-SNP heritability of MH SNPs $\widehat{\Delta}_{h^2}$ while controlling for age, sex, the first 20 genetic PCs estimated from common SNPs, and the first five PCs estimated from NIMs (NIM PCs). The five NIM PCs are computed using all NIMs in unrelated white British samples with ProPCA (**Agrawal et al., 2020**).

To improve power to detect patterns that are shared across groups of phenotypes, we combined analyses across groups of phenotypes and across all phenotypes analyzed. We performed random

effect meta-analysis on each phenotypic category containing at least four phenotypes. We assume that the phenotypes within each category $i$ have their $\widehat{h^2}_{NIM}$ drawn from the same distribution so that we can estimate the mean ($meta - h^2_{NIM}$) and variance of distribution $i$, based on the sampled $\widehat{h^2}_{NIM}$ and the s.e.($\widehat{h^2}_{NIM}$). From there, we computed the meta analysis Z-score to test if the $meta - h^2_{NIM}$ is equal to zero. Similarly, we assume the phenotypes within each category $i$ have their $\Delta_{h^2}$ drawn from the same distribution, and compute the Z-score to test if the $meta - \Delta_{h_2}$ is equal to zero. In addition to the meta-analysis within the phenotypic category, we also performed meta-analysis across all phenotypes where we used a subset of 32 phenotypes that were chosen to have low correlation (Pearson's $r^2 \leq 0.25$).

## Identifying individual NIMs associated with phenotype

To identify individual NIMs associated with a phenotype, we fit a linear regression model using plink 2.0 `--glm` and include covariates controlling for age, sex, and the first 20 genotypic PCs, and first five NIM PCs. We used a stringent p-value threshold of $10^{-10}$ to correct for the number of NIMs and phenotypes tested. For each phenotype, we clumped all significant NIMs that lie within 250 kb and with an LD threshold ($r^2$) of 0.5 using a significance threshold for the index SNP of $10^{-10}$.

## Identifying NIMs that modulate phenotype

To assess our ability to identify introgressed variants that truly modulate a phenotype, we first tested each NIM for association with the simulated phenotype. A challenge with such an approach is the possibility that an NIM can be found to be associated with a phenotype due to being in LD with a non-introgressed variant. To exclude settings where the association signal at an NIM might be driven by LD with a non-introgressed variant, we applied a Bayesian statistical fine-mapping method (SuSiE, **Wang et al., 2020**) that analyzes both NIM and MH SNPs in the region surrounding an associated NIM to output a set of SNPs that can explain the association signal at the region. Furthermore, we processed these credible sets to obtain a set of credible NIMs.

We performed simulations to test the accuracy of such an approach in identifying truly causal NIMs. In particular, we first ran an association test with plink (https://www.cog-genomics.org/plink/) to identify significant NIMs (p-value $<10^{-10}$). We then LD-pruned significant NIMs to get a subset of NIMs that are approximately uncorrelated with each other (using the plink flag "--indep-pairwise 100 kb 1 0.99"). For each LD-pruned significant NIM, we considered all the QC-ed SNPs in its 200 kb neighborhood as input to fine mapping. We ran SuSiE with ρ = 0.95 and $L$ = 10, such that it returns credible sets that have at least 0.95 probability to contain one causal variant and outputs at most 10 credible sets for each tested region. If there are more than one credible set for a tested region, we merge them into one set. We then removed the credible sets which have 50% or more MH SNPs in their credible set. The remaining credible sets all have majority NIMs (i.e., positive results), and they are further merged together with other such regions it overlaps with, resulting in distinct regions with evidence of NIM causal effects. We termed the set of all resulting NIMs as the credible NIM set and all NIMs that lie in the credible set as credible NIMs. The region containing the credible NIM set is termed credible NIM region. If there is at least one true causal NIM within the set of credible NIMs, this credible NIM region is counted as a true positive (TP). If there is no causal NIM in the credible NIMs, this credible NIM region is counted as a false positive (FP).

We adopted the same approach when analyzing UKBB phenotypes while incorporating covariates. Because the SuSiE package does not directly incorporate covariates, we used regression residuals from linear regression between each UKBB phenotype and UKBB covariates (age, sex, 20 regular PCs, 5 NIM PCs), as the input phenotype to SuSiE.

## Annotating NIMs

We annotated all unique credible NIMs using SnpEff (**Cingolani et al., 2012**) that uses Sequence Ontology (http://www.sequenceontology.org/) to assign standardized terminology for assessing sequence change and impact. We primarily focused on examining the high (e.g., start codon loss, stop codon gain) and moderate impact SNPs (nonsynonymous variants) that are coding variants that alter protein sequences. We used FUMA (**Watanabe et al., 2017**) to annotate the unique credible NIMs as eQTLs (https://fuma.ctglab.nl/). To analyze NIM heritability together with a measure of background selection, we annotated NIMs with the B-value (**McVicker et al., 2009**), a measure of background

selection (https://github.com/gmcvicker/bkgd, copy archived at swh:1:rev:5251f317b2261e06ad-ba58fd454d41710079d3b5; *McVicker, 2020*).

## Acknowledgements

We thank Bogdan Pasaniuc and Kirk Lohmueller for feedback on the manuscript. DR was supported by an Allen Discovery Center grant on Brain Evolution from the Paul Allen Foundation, John Templeton Foundation grant 61220, US National Science Foundation HOMINID grant BCS-1032255, US National Institutes of Health grants GM100233 and HG006399, and is an Investigator of the Howard Hughes Medical Institute. SS is supported in part by NIH grants R35GM125055, NSF grants III-1705121, CAREER-1943497, an Alfred P Sloan Research Fellowship, and a gift from the Okawa Foundation. PL was supported by a Burroughs Wellcome Fund Career Award at the Scientific Interfaces and the Next Generation Fund at the Broad Institute of MIT and Harvard.

This research was conducted using the UK Biobank Resource under applications 12408 and 33127.

## Additional information

### Funding

| Funder | Grant reference number | Author |
|---|---|---|
| National Institutes of Health | GM100233 | David Reich |
| National Science Foundation | BCS1032255 | David Reich |
| National Institutes of Health | HG006399 | David Reich |
| Paul G. Allen Frontiers Group | Allen Discovery Center Grant on Brain Evolution | David Reich |
| John Templeton Foundation | 61220 | David Reich |
| Howard Hughes Medical Institute | | David Reich |
| National Institutes of Health | R35GM125055 | Sriram Sankararaman |
| National Science Foundation | III1705121 | Sriram Sankararaman |
| National Science Foundation | CAREER1943497 | Sriram Sankararaman |
| Alfred P. Sloan Foundation | | Sriram Sankararaman |
| Okawa Foundation | | Sriram Sankararaman |
| Burroughs Wellcome Fund | | Po-Ru Loh |
| Next Generation Fund at the Broad Institute of MIT and Harvard | | Po-Ru Loh |

The funders had no role in study design, data collection and interpretation, or the decision to submit the work for publication.

### Author contributions

Xinzhu Wei, Data curation, Software, Formal analysis, Validation, Investigation, Visualization, Methodology, Writing - original draft, Writing – review and editing; Christopher R Robles, Data curation, Formal analysis, Validation, Investigation, Visualization, Writing - original draft, Writing – review and editing; Ali Pazokitoroudi, Data curation, Software, Investigation, Methodology; Andrea Ganna, Data curation, Investigation, Writing – review and editing; Alexander Gusev, Arun Durvasula, Steven Gazal,

Po-Ru Loh, Investigation, Writing – review and editing; David Reich, Conceptualization, Investigation, Writing – review and editing; Sriram Sankararaman, Conceptualization, Resources, Data curation, Formal analysis, Supervision, Funding acquisition, Investigation, Visualization, Methodology, Writing - original draft, Project administration, Writing – review and editing

**Author ORCIDs**
Xinzhu Wei ⓘ http://orcid.org/0000-0001-8184-7016
Christopher R Robles ⓘ http://orcid.org/0000-0001-5667-7625
Ali Pazokitoroudi ⓘ http://orcid.org/0000-0002-2839-2291
Arun Durvasula ⓘ http://orcid.org/0000-0003-0631-3238
David Reich ⓘ http://orcid.org/0000-0002-7037-5292
Sriram Sankararaman ⓘ http://orcid.org/0000-0003-1586-9641

**Decision letter and Author response**
Decision letter https://doi.org/10.7554/eLife.80757.sa1
Author response https://doi.org/10.7554/eLife.80757.sa2

## Additional files

**Supplementary files**
• MDAR checklist

**Data availability**
New software is deposited on GitHub: https://github.com/alipazokit/RHEmc-coeff, (copy archived at swh:1:rev:b53cfba3f8f8dd160082dda642075302f64d46a0). Data for figures (and supplement data) are deposited on GitHub: https://github.com/AprilWei001/NIM, (copy archived at swh:1:rev:16f-15da80f182ccc4dcab3a9993819372beedd5c). Supplementary data includes: Data S1 - UKBB phenotype annotation; Data S2 - RHE-mc results in simulated data; Data S3 - RHE-mc results with Ancestry+MAF+LD annotations and NIM PCs included in covariates applied to 96 UKBB phenotypes; Data S4 - RHE-mc results with Ancestry only annotation and NIM PCs included in covariates applied to 96 UKBB phenotypes; Data S5 - RHE-mc results with Ancestry+MAF+LD annotation without NIM PC in covariates applied to 96 UKBB phenotypes; Data S6 - Fine mapping FDP in simulated data; Data S7 - 112 credible NIM sets and credible NIMs; Data S8 - SnpEff annotation of all unique credible NIMs; Data S9 - Stratified LD score regression results in simulated data using LD score from 1KG; Data S10 - Stratified LD score regression results in simulated data using LD score from UKBB.

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

# Appendix 1

## Identification of SNPs that tag Neanderthal ancestry on the UK Biobank Axiom array

Starting with the confidently inferred Neanderthal haplotypes identified in *Sankararaman et al., 2014*, we identified whether an SNP segregating in a target modern human population owes its origin to the Neanderthal gene flow event as follows:

1. We identified sets of haplotypes that are confidently labeled as Neanderthal, *N* by the conditional random field (CRF) method proposed in *Sankararaman et al., 2014* scanning for runs of consecutive SNPs with marginal probability of Neanderthal ancestry ≥0.90. We also identified sets of haplotypes that are confidently labeled as non-Neanderthal, *MH* by scanning for SNPs with marginal probability ≤0.1. We also required the Neanderthal haplotype to be at least 0.02 cM long.
2. For each SNP called in the 1000 Genomes dataset, we required that none of the derived alleles at this SNP falls on one of the modern human haplotypes in the set *MH* and all of the haplotypes in *N* carry the derived allele. This procedure allows for some false negatives in the predictions of the CRF.
3. We ran this procedure on the combined calls from the European ancestry populations (CEU, GBR, FIN, IBS, and TSI) in the 1000 Genomes Project.

This procedure yielded a total of 95,462 SNPs that are likely to be Neanderthal-derived. We winnowed down this list to 43,026 SNPs after removing ones already tagged at $r^2 > 0.8$ by SNPs on the UKBiLEVE array. We then designed a greedy algorithm to capture the remaining untagged SNPs that could still be accommodated on the array (we determined the number of oligonucleotide features that would be needed to genotype each SNP as well as the total number of features available on the array through discussions with UK Biobank Axiom array design team).

Specifically, we computed LD between all pairs of Neanderthal-derived SNPs and then iteratively picked SNPs with the highest score to add to the array where the score was computed as

$$Score_{SNP\ j} = \frac{\sum_{i=1}^{n} \left[ \delta_{r^2 > 0.80} (i,j) \right] \left[ Derived\ frequency_{SNP\ i} \right]}{Features\ required\ genotype\ SNP\ j}$$

Here $\delta_{r^2 > 0.80} (i,j)$ is an indicator variable that is 1 if the squared correlation coefficient between SNPs $i$ and $j$ is >0.80 and zero otherwise. Thus, SNP j is scored higher if it tags other untagged SNPs on the array. The other two terms upweight SNPs that tag other SNPs with high-derived allele frequency in Europeans and downweight SNPs by the number of oligonucleotide features required to genotype it.

We iteratively chose SNPs until we obtained 6,027 SNPs (requiring 16,674 features) that fully tagged the remaining set of Neanderthal-derived SNPs. These 6027 SNPs were then added to the UK Biobank Axiom array.

## Appendix 2

### Estimating NIM heritability with partitioned LD-score regression

We considered two candidate methods for estimating the NIM heritability in large datasets and testing the related hypotheses to NIM heritability, S-LDSR (*Finucane et al., 2015*) and RHE-mc (see main text) (*Pazokitoroudi et al., 2020*). S-LDSR can speedily estimate partitioned heritability given GWAS statistics and LD scores without any individual-level data. S-LDSR can be used with either in-sample LD scores (i.e., computed from the same data as for GWAS) or out-of sample LD scores (i.e., computed from an external and often much smaller data set). Out-of-sample LD scores from 1000 Genomes (1KG) is often used in S-LDSR (*McArthur et al., 2021*; *Koller et al., 2021*) because (1) it is computationally much cheaper to compute than using the GWAS cohorts, and (2) individual-level data from GWAS cohorts are not always accessible; despite that, S-LDSR with in-sample LD scores is more accurate in theory.

Previous studies by Koller et al. and McArthur et al. used S-LDSR to estimate the heritability from archaic ancestries. They computed the stratified LD scores using the 1000 Genomes (1KG) EUR and EAS samples and performed LD score regression against the GWAS statistics from a different cohort. If the ancestry from 1KG samples does not match well with the GWAS cohort, it could lead to biased heritability estimates. Additionally, the LD score distribution and MAF distributions of NIMs are very different from the distributions of MH SNPs (*Figure 1*), which might also affect the heritability estimates if not taken into account. Finally, LD score regression is restricted to a subset of SNPs (typically with MAF >5%), which substantially reduces the number of NIMs analyzed. Here, we benchmarked S-LDSR on the simulated data with both out-of-sample LD scores from 1KG and the in-sample LD scores from all UKBB QC-ed data, stratified by ancestry (NIM vs. MH).

First, we used the aforementioned simulations to evaluate the partitioned LD score regression in estimating NIM heritability. We downloaded the 1KG EUR data (from https://storage.googleapis.com/broad-alkesgroup-public/LDSCORE/1000G_Phase3_plinkfiles.tgz) that is typically used for LD score regression. There are 9,997,231 SNPs in the data, and 5,789,471 of them are shared with the UKBB QC-ed SNPs. Out of the 235,592 expanded NIMs defined in UKBB QC-ed data, 210,962 are present in 1KG EUR, and we refer to these SNPs as the 1KG NIMs. We defined the 9,786,269 SNPs in 1KG EUR that are not expanded NIMs as 1KG MH SNPs. We then computed the stratified LD score using GCTA software with flags `--ld-score --ld-wind 10000` for all the 1KG SNPs, all the 1KG NIMs, and all the 1KG MH SNPs. From here, we computed the stratified LD score with $ldsc_{SNP} = ldsc_{NIM} + ldsc_{MH}$ , such that each SNP has two stratified LD scores, one due to its LD with NIMs and another due to its LD with MH SNPs. We then intersected the 1KG SNPs with UKBB QC-ed SNPs, and used the shared 5,789,471 SNPs to perform stratified LD score regression. The S-LDSR is then performed with all SNPs that overlap between 1KG and UKBB. For each simulation, we ran S-LDSR to estimate $h^2_{NIM}$ , $h^2_{MH}$ , $\Delta_{h^2}$ , and their standard errors from 200 jackknife blocks. We found that the results from using out-of-sample LD are biased even when heritability does not depend on MAF and LD (i.e., BASELINE) (*Appendix 2—figure 1*).

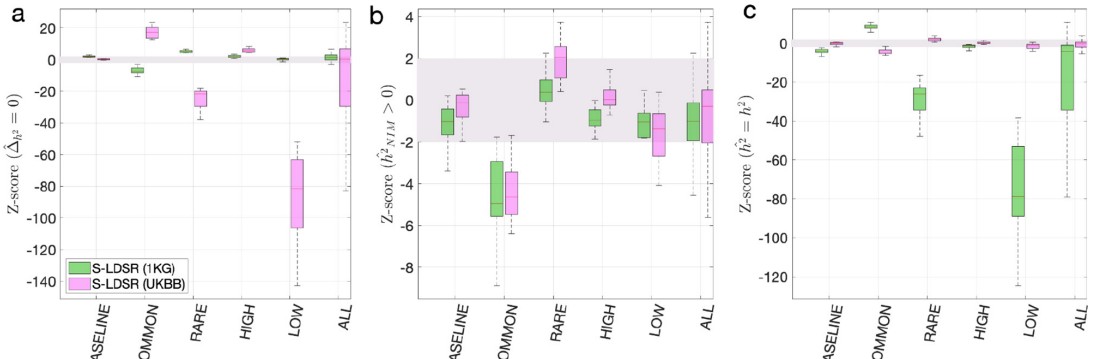

**Appendix 2—figure 1.** Benchmark stratified LDSC regression (S-LDSR) with in-sample and out-of-sample LD scores. We group the simulations by the MAF-LD coupling: BASELINE, COMMON, RARE, HIGH, LOW, and ALL, as labeled on the x-axis. We plot the distributions of three Z-scores (y-axis), one on each panel: (**a**) Z-score $(\hat{\Delta}_{h^2} = 0)$ tests whether the estimated Neanderthal Informative Mutation (NIM) heritability is different from the

*Appendix 2—figure 1 continued*

matched modern human (MH) heritability, (**b**) Z-score ($\widehat{h^2}_{NIM} = h^2_{NIM}$) tests whether the estimated and expected NIM heritability are equal, and (**c**) Z-score ($\widehat{h^2} = h^2$) tests whether the estimated and simulated total heritability are equal. In each panel, S-LDSR with the out-of-sample LD score from 1000 Genomes (1KG) is shown in green and S-LDSR with in-sample LD score from UKBB in pink. In S-LDSR, only ancestry annotation is used. The Z-scores within ±2 are color shaded. S-LDSR (1KG) is not calibrated even for BASELINE architecture.

As a comparison, we computed the in-sample stratified LD score using the UKBB QC-ed data and applied S-LDSR with these in-sample LD scores. In contrast to the previous results, the results are well calibrated for BASELINE, suggesting that the previous biases observed with BASELINE are due to the disagreement between the out-of-sample LD score and the in-sample LD score (*Appendix 2— figure 1*). Not surprisingly, the results for MAF and LD-dependent architectures are still biased as these factors are not taken into account. We caution that our simulations are based on UKBB QC-ed SNPs, where non QC-ed SNPs do not have an impact on the simulated phenotypes. This setting will favor S-LDSR based on UKBB QC-ed SNPs more than in actual settings, and disfavor S-LDSR 1KG more than in actual settings. It is possible that in reality the biases with in-sample LD score will become larger, and the biases with out-of-sample LD score will become smaller. Nonetheless, because it is often expensive to compute in-sample LD scores, the accuracy will largely depend on how well the external panel resembles the GWAS cohort.

The out-of-sample LD score could be particularly biased for low MAF SNPs, hence S-LDSR recommends not using annotations with fewer than 5% of SNPs as best practice. This practice will necessarily exclude more than 70% of NIMs and about half of the MH SNPs, and the heritability estimates from high MAF SNPs may not extrapolate to low MAF SNPs. Therefore, S-LDSR, under the best practice, is not suitable for studying Neanderthal introgressed variants.

## Appendix 3

### The impact of inclusion of NIM PCs on NIM heritability estimates

We computed the first five NIM PCs using all NIMs in unrelated white British samples with ProPCA (*Agrawal et al., 2020*). Compared to the regular genetic PCs (estimated from common SNPs), NIM PCs are only weakly correlated with birth GPS locations (*Appendix 3—figure 1*), consistent with the fact that Neanderthal introgression occurred soon after the out-of-African migration before population expansion.

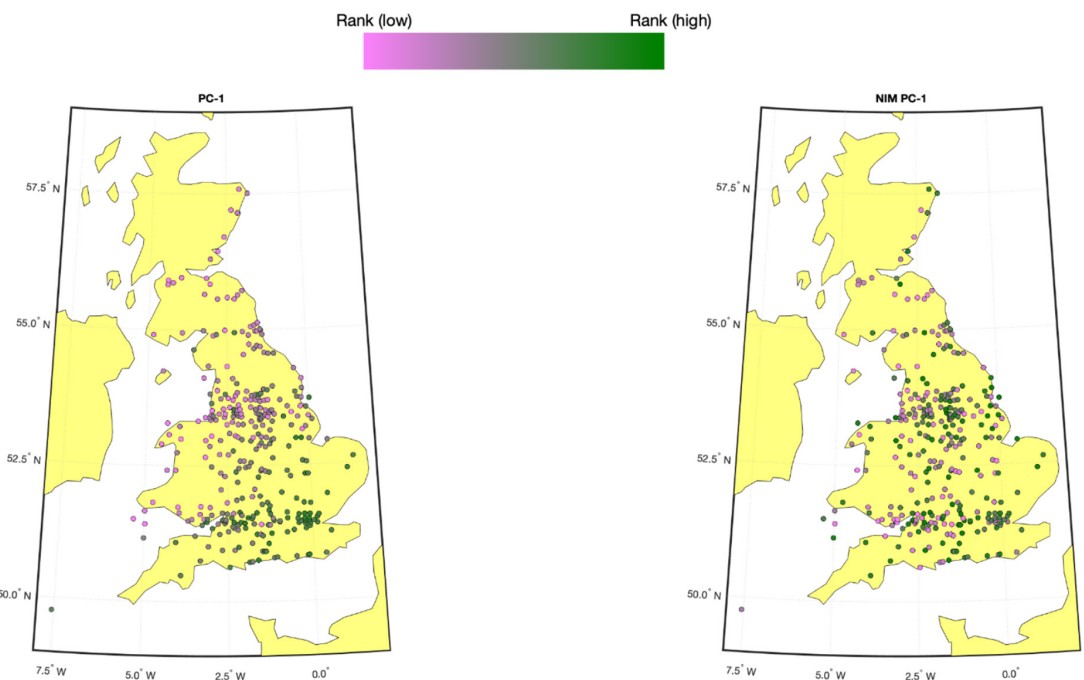

**Appendix 3—figure 1.** Population structure within white British samples. PC-1 from the whole genome genotypes (released by UK Biobank [UKBB]) is shown on the left, and Neanderthal Informative Mutation (NIM) PC-1 is shown on the right. We used a 20-by-20 grid along the latitude and longitude, dividing the map into 400 colonies. We then computed the average PC projection as well as the median longitude and latitude among the individuals belonging to each colony, if there are at least 10 individuals in a colony. Each color-filled circle with a 5 km radius represents one colony on the map. To maximize the visible differences, we sorted the colonies by their PC values and used the rank to determine the color of the colony. Compared to NIM PC-1, PC-1 shows a much stronger correlation with geographical location.

When NIM PCs were not being controlled for (with remaining regular covariates still used), we found three phenotypes with significant NIM heritability (Z-score ($\widehat{h^2}_{NIM} = 0$) > 3): overall health rating, waist-hip-ratio (WHR), and gamma glutamyltransferase (a measure of liver function). We also combined phenotypes into broader phenotypic categories and performed random effect meta-analysis on the nine categories that contain at least four phenotypes (see 'Methods'). We found that $meta - \widehat{h^2}_{NIM}$ is significantly larger than zero (Z-score > 2.53 for one-tail p=0.05 level) for all but two categories (eye, lipid metabolism), meaning that NIMs heritability is generally nonzero (*Appendix 3—figure 2a and c*). We then tested whether NIM heritability is larger or smaller compared to MH SNPs ($\hat{\Delta}_{h^2} = 0$). Fourteen phenotypes (standing height, sitting height, weight, body fat percentage, whole-body fat-free mass, whole-body water mass, whole-body impedance, trunk fat-free mass, trunk predicted mass, basal metabolic rate, RBC count, apolipoprotein A, HDL cholesterol, triglycerides) remain significantly depleted (Z-score < –3), among which 10 are anthropometric phenotypes, and 3 are related to lipid metabolism. This is in contrast to 17 phenotypes when NIM PCs are controlled for (body mass index, hip circumference, waist circumference, standing height, sitting height, weight, whole-body fat-free mass, whole-body water mass, whole-body impedance, trunk fat mass, trunk fat-free mass, trunk predicted mass, basal metabolic rate, RBC count, apolipoprotein A, HDL cholesterol, triglycerides).

Five phenotypic categories show significant NIM heritability depletion (anthropometry, blood biochemistry, blood pressure, lipid metabolism, lung), and four are not significantly different with meta analysis (*Appendix 3—figure 2b and d*). In contrast to the evidence for depletion in NIM heritability, we found no evidence for traits with elevated NIM heritability even when excluding NIM PCs (*Appendix 3—figure 2d*).

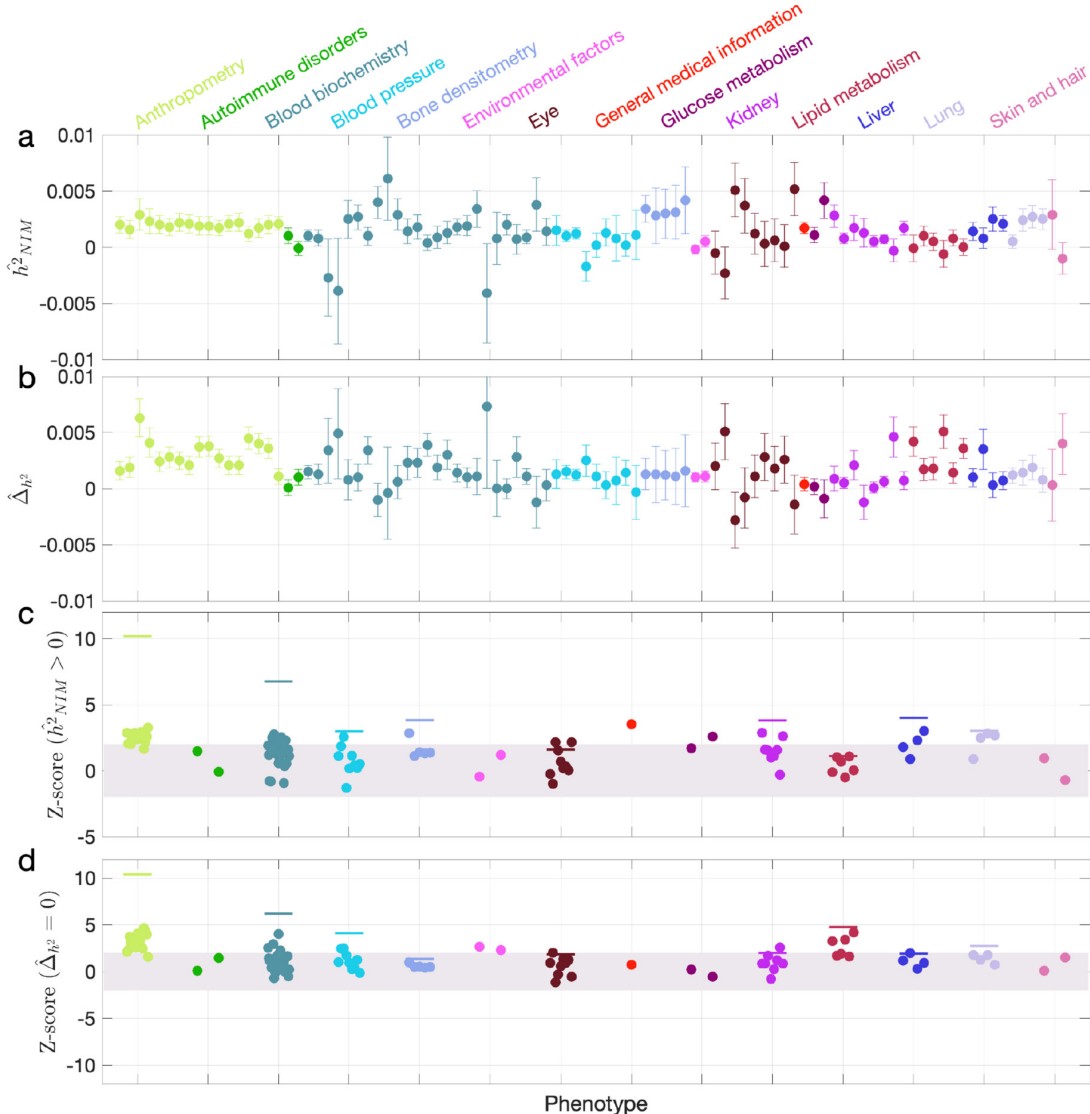

**Appendix 3—figure 2.** Neanderthal Informative Mutation (NIM) heritability in the 96 UK Biobank (UKBB) phenotypes without controlling for NIM principal components (PCs). This figure is plotted in the same way as *Figure 3*. Heritability estimates are largely similar, but fewer phenotypes are significant. Three phenotypes have significant positive NIM heritability (Z-score ($\widehat{h^2}_{NIM} = 0$)>3): overall health rating, waist-hip-ratio, and gamma glutamyltransferase. Fourteen phenotypes (standing height, sitting height, weight, body fat percentage, whole-body fat-free mass, whole-body water mass, trunk fat-free mass, trunk predicted mass, basal metabolic rate, RBC count, apolipoprotein A, HDL cholesterol, triglycerides) are significantly depleted for NIM heritability (Z-score < −3).

## Appendix 4

## Statistic to compare per-NIM heritability to per-SNP heritability at a set of background MH SNPs

In this note, we provide additional intuition behind our statistic to compare difference between the heritability at an NIM (per-NIM heritability) and the per-SNP heritability of a background set of MH SNPs:

$$\hat{\Delta}_{h^2} = \widehat{h^2}_{NIM} - \widehat{h^2}_{MH}$$

Let $\sigma^2_{a,i} = \frac{h^2_{a,i}}{M_{a,i}}$ where $a \in \{NIM, MH\}$, $i$ denotes one of the annotations (MAF,LD), $h^2_{a,i}$ denotes the heritability attributed to annotation $(a,i)$, and $M_{a,i}$ denotes the number of SNPs in annotation $(a,i)$. Thus $\sigma^2_{a,i}$ denotes the per-SNP heritability associated with annotation $(a,i)$.

The per-SNP heritability associated with NIMs is given by

$$\sigma^2_{NIM} = \sum_i \frac{\sigma^2_{NIM,i} M_{NIM,i}}{M_{NIM}} = \sum_i \frac{h^2_{NIM,i}}{M_{NIM}} = \frac{1}{M_{NIM}} \sum_i h^2_{NIM,i}$$

where $M_{NIM}$ denotes the total number of NIMs.

To choose a background set of MH SNPs that match the NIMs in terms of their MAF and LD distribution, we would pick a given bin $i$ with probability $\frac{M_{NIM,i}}{M_{NIM}}$. The per-SNP heritability associated with this background set of MH SNPs is then given by

$$\sigma^2_{MH} = \sum_i \frac{\sigma^2_{MH,i} M_{NIM,i}}{M_{NIM}} = \frac{1}{M_{NIM}} \sum_i h^2_{MH,i} \frac{M_{NIM,i}}{M_{MH,i}}$$

Thus, we are interested in testing the null hypothesis that the per-NIM heritability is equal to the per-SNP heritability of the background set of MH SNPs.

$$\sigma^2_{NIM} - \sigma^2_{MH} = 0$$

$$\Rightarrow \frac{1}{M_{NIM}} \sum_i h^2_{NIM,i} - \frac{1}{M_{NIM}} \sum_i h^2_{MH,i} \frac{M_{NIM,i}}{M_{MH,i}} = 0$$

$$\Rightarrow \sum_i h^2_{NIM,i} - \sum_i \frac{M_{NIM,i}}{M_{MH,i}} h^2_{MH,i} = 0$$

Defining our parameter of interest $\Delta_{h^2} = \sum_i h^2_{NIM,i} - \sum_i \frac{M_{NIM,i}}{M_{MH,i}} h^2_{MH,i}$, our null hypothesis is that $\Delta_{h^2} = 0$. We estimate the relative reduction in NIM heritability as

$$\delta_{h^2} = \frac{\sigma^2_{NIM} - \sigma^2_{MH}}{\sigma^2_{MH}}$$

$$= \frac{\sum_i h^2_{NIM,i} - \sum_i \frac{M_{NIM,i}}{M_{MH,i}} h^2_{MH,i}}{\sum_i \frac{M_{NIM,i}}{M_{MH,i}} h^2_{MH,i}}$$

## Appendix 5

### Estimating NIM heritability with NIMs defined by Sprime

To assess the robustness of our results to the methodology used to identify NIMs, we performed additional analyses with NIMs annotated by Sprime. We used two sets of NIMs that had been identified by Sprime in the 1000 Genomes Project (1KG) and were analyzed by *McArthur et al., 2021*: the least stringent set of 900,902 putatively introgressed variants identified in 1KG subpopulations regardless of evidence of matching the Neanderthal allele and the most stringent set of 138,774 putatively introgressed variants identified in the 1KG European subpopulations matching the Altai Neanderthal allele. We annotated the ancestry of the QC-ed SNPs in UKBB with the two sets of NIMs identified from Sprime and analyzed the heritability of the 96 UKB phenotypes at these NIMs (accounting for MAF and LD).

We observed that both sets of NIMs overlap with the set of NIMs that were identified using the CRF with 73,675 NIMs that are present in the most stringent set and 166,756 that are present in the least stringent set. We compared the MAF and LD scores at NIMs to MH SNPs at both sets of SNPs and found that they have similar MAF and LD distributions as the expanded NIMs (*Appendix 5—figure 1* and *Appendix 5—figure 2*). We then tested whether NIM heritability at these NIMs is larger or smaller compared to MH SNPs ($\hat{\Delta}_{h^2} = 0$). Consistent with our findings based on NIMs identified using the CRF, we did not identify an enrichment in heritability across any of the traits (*Appendix 5—figure 3* and *Appendix 5—figure 4*). Eighteen phenotypes were significantly depleted when analyzing the least stringent set while two were significantly depleted in the more stringent set which we hypothesize is due to the smaller number of SNPs in this set.

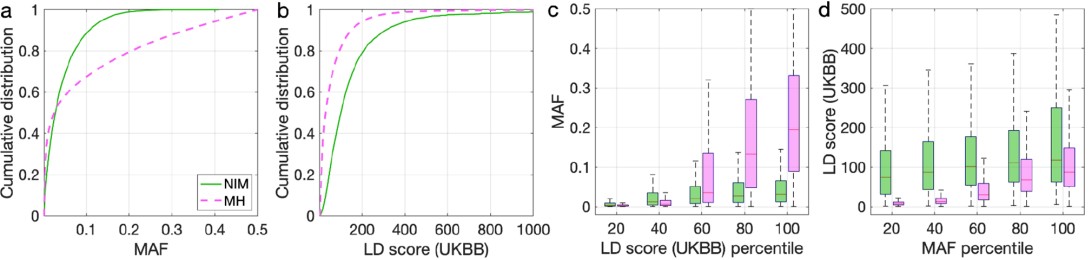

**Appendix 5—figure 1.** Distributions of minor allele frequency (MAF) and LD-score in Neanderthal Informative Mutations (NIMs) identified by Sprime in all 1KG populations and modern human (MH) SNPs. Empirical cumulative distribution functions of (**a**) MAF and (**b**) LD scores of NIMs identified by Sprime (in solid green line) and MH SNPs (in pink dashed line) estimated in the UK Biobank (UKBB). (**c**) Boxplots of MAFs of NIMs (on the left filled in green) and MH SNPs (on the right side filled in pink) while controlling for LD score (UKBB). (**d**) Boxplots of LD score (UKBB) of NIMs and MH SNPs while controlling for MAF. NIMs and MH SNPs are divided by the 20, 40, 60, 80, 100 (**c**) LD score (UKBB) percentile or MAF percentile (**d**) based on all QC-ed SNPs (7,774,235 imputed SNPs with MAF > 0.001). The lower and upper edges of a box represent the first and third quartile (qu1 and qu3), respectively; the horizontal red line inside the box indicates median (md); the whiskers extend to the most extreme values inside inner fences, md ± 1.5 (qu3–qu1).

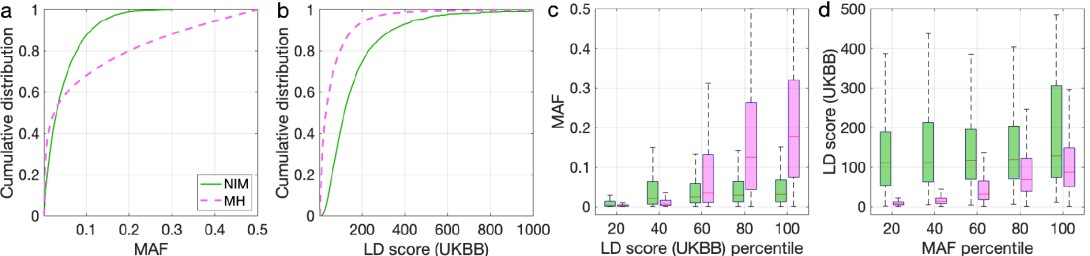

**Appendix 5—figure 2.** Distributions of minor allele frequency (MAF) and LD-score in Neanderthal Informative Mutations (NIMs) identified by Sprime in 1KG European populations and modern human (MH) SNPs. Empirical cumulative distribution functions of (**a**) MAF and (**b**) LD scores of NIMs identified by Sprime (in solid green line) and MH SNPs (in pink dashed line) estimated in the UK Biobank (UKBB). (**c**) Boxplots of MAFs of NIMs (on the left

*Appendix 5—figure 2 continued*
filled in green) and MH SNPs (on the right side filled in pink) while controlling for LD score (UKBB). (**d**) Boxplots of LD score (UKBB) of NIMs and MH SNPs while controlling for MAF. NIMs and MH SNPs are divided by the 20, 40, 60, 80, 100 (**c**) LD score (UKBB) percentile or MAF percentile (**d**) based on all QC-ed SNPs (7,774,235 imputed SNPs with MAF > 0.001). The lower and upper edges of a box represent the first and third quartile (qu1 and qu3), respectively; the horizontal red line inside the box indicates median (md); the whiskers extend to the most extreme values inside inner fences, md ± 1.5 (qu3–qu1).

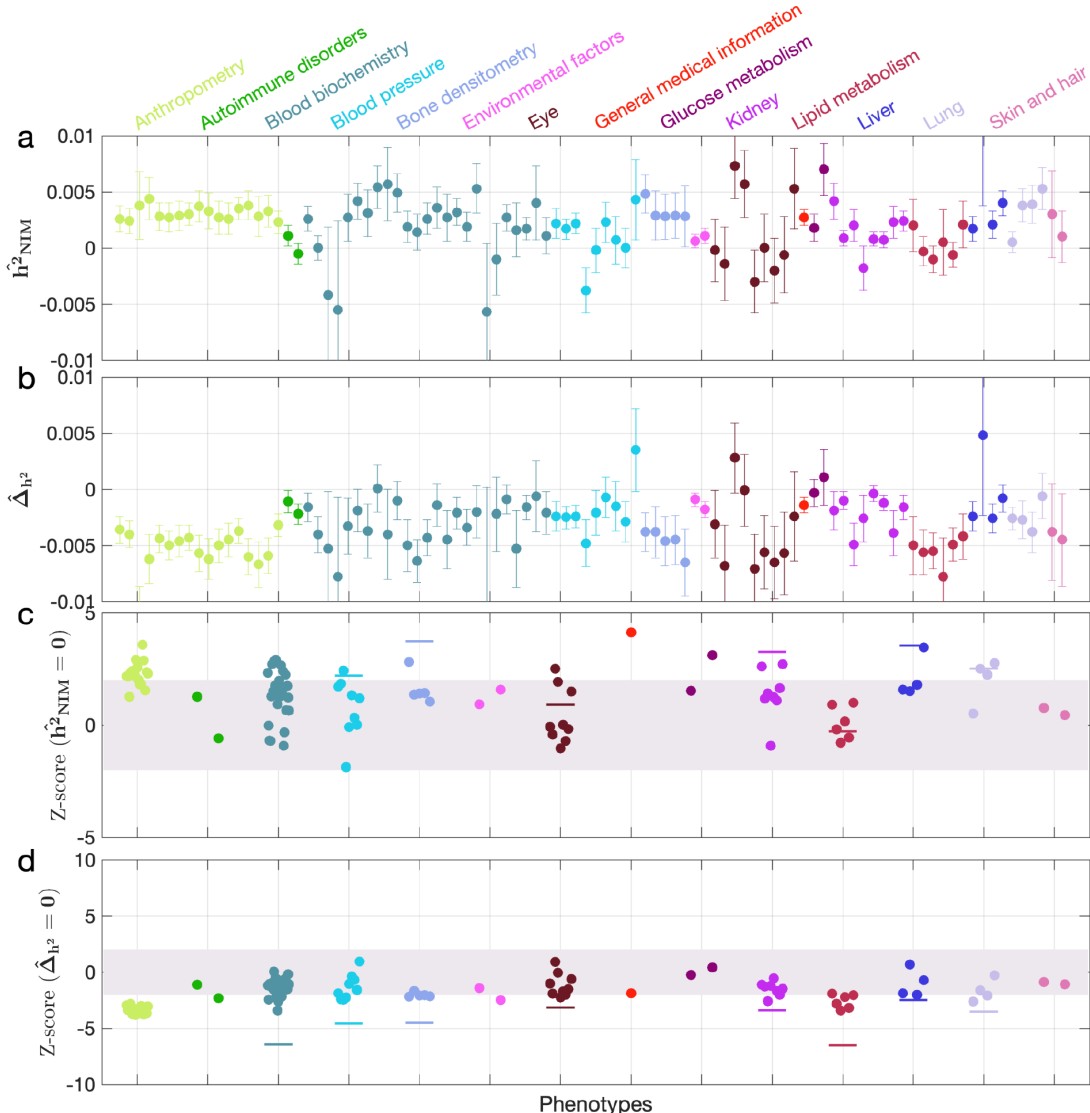

**Appendix 5—figure 3.** Neanderthal Informative Mutation (NIM) heritability in the 96 UK Biobank (UKBB) phenotypes for the least stringent set of NIMs identified using Sprime by *McArthur et al., 2021*. (**a**) Estimates of NIM heritability ($\widehat{h^2}_{NIM}$) and (**c**) the Z-score of $\widehat{h^2}_{NIM}$ (testing the hypothesis that NIM heritability is positive) for each UKBB phenotype. Analogously, (**b**) estimates of $\hat{\Delta}_{h^2}$ and Z-score (**d**) of $\hat{\Delta}_{h^2}$ (testing the hypothesis that per-NIM heritability is equal to per-SNP heritability at modern human [MH] SNPs after controlling for MAF and LD).

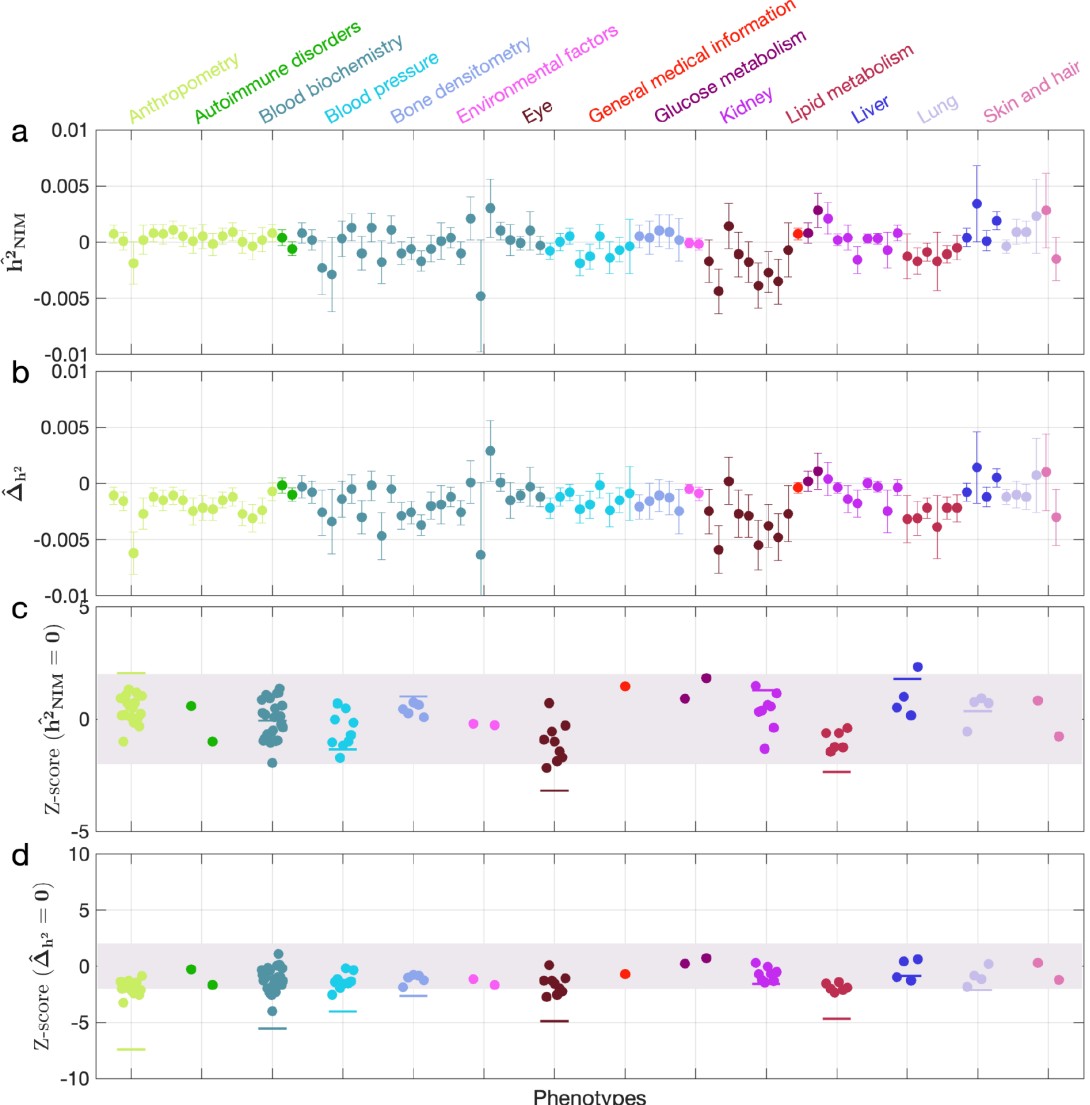

**Appendix 5—figure 4.** Neanderthal Informative Mutation (NIM) heritability in the 96 UK Biobank (UKBB) phenotypes for the most stringent set of NIMs identified using Sprime by *McArthur et al., 2021* NIMs. (**a**) Estimates of NIM heritability ($\widehat{h^2}_{NIM}$) and (**c**) the Z-score of $\widehat{h^2}_{NIM}$ (testing the hypothesis that NIM heritability is positive) for each UKBB phenotype. Analogously, (**b**) estimates of $\hat{\Delta}_{h^2}$ and Z-score (**d**) of $\hat{\Delta}_{h^2}$ (testing the hypothesis that per-NIM heritability is equal to per-SNP heritability at modern human [MH] SNPs after controlling for MAF and LD).

