## [Editor Report]

Humans whose genetic ancestors lived outside Africa have a small proportion of the genome that traces back to interbreeding events with Neanderthals. To quantify the contribution of this ancestry to present-day phenotypic variation, the authors develop a convincing set of approaches that takes into account various complicating factors and apply it to a subset of the UK Biobank individuals. The work is an important contribution to human evolution and evolutionary biology more generally.

---

## [Decision Letter]

**Decision letter after peer review:**

Thank you for submitting your article "The lingering effects of Neanderthal introgression on human complex traits" for consideration by *eLife*. Your article has been reviewed by 2 peer reviewers, and the evaluation has been overseen by a Reviewing Editor and Molly Przeworski as the Senior Editor. The following individual involved in review of your submission has agreed to reveal their identity: Lawrence Uricchio (Reviewer #2).

The reviewers have discussed their reviews with one another. The full reviews are below and give a set of reasonable suggestions to improve the paper. In discussions with the reviewers, the importance of clarifying the points of agreement with past work came through clearly.

Reviewing Editor comments:

I enjoyed this paper. I include below a comment on top of those provided by the reviewers.

In addition to matching to modern human SNPs for LD and MAF, it would be good to match on the basis of B-value (or a related metric). Loci with still polymorphic Neanderthal alleles are likely to be further from genes and in higher recombination regions. Doing this would allow us to move somewhat towards the conclusion that particular traits were under selection against Neanderthal alleles and not just selection against functional changes in general. The latter would still be of interest, but it's good to think about layered null models.

*Reviewer #1 (Recommendations for the authors):*

In the section describing the whole-genome simulations of traits, it would help to clarify how the effect sizes are drawn. In particular, are the β values independent of frequency, but the per-SNP effects are normalized such that h^2 does not vary with frequency? This has been done in some previous studies for statistical convenience, and it has been argued that this type of model is qualitatively consistent with weak negative selection. In my opinion it is not a particularly plausible model for an effect size distribution, and when we have tried simulating negative selection models the joint distribution of effect sizes and frequencies can deviate widely from such procedures. Still, so long as the procedure is clarified I think it is fine for the purposes of this study.

In general, I'm of the opinion that the best way to assess a method such as yours would be to perform forward in time simulations that include selection and demography and to assess the method on simulated traits, such that the joint distributions of frequencies/LD/effects is determined by the model. However, I recognize that this would be a large project and that you have made substantial effort to assess the method rigorously already. I also agree with what you said at the end of your paper, that the space of models that could correspond with your results is still quite vast, meaning that such simulations would necessarily focus on a small subset of potential models. I do think that adding some simulations with more extreme contributions due to rare variants would be helpful to establish the applicability of the method here.

*Reviewer #2 (Recommendations for the authors):*

I have 2 suggestions for analyses that would make this study stronger:

1) This work makes an important point that future studies of introgressed haplotypes should account more carefully about MAF and LD patterns in their study populations of interest. Another choice such studies have to make is which set of putative introgressed haplotypes to use; there are several choices available that use LD information in different ways. It would be useful to know how robust these patterns are to differences in regions identified by those methods. It would also be helpful for interpreting differences between this and previous heritability studies; both McArthur 2021 and the Koller 2021 preprint cited in the appendix rely on the Sprime method of identifying introgressed haplotypes (Browning et al. 2018) rather than the CRF-based method used here. It would be informative to know at least how many haplotypes considered here are identified by the other methods, and whether they have the same distribution of MAF and LD score, and same general heritability trends.

2) Focusing on coding changes is a reasonable starting point given their likelihood of affecting proteins. However, given the prevalence of noncoding changes associated with phenotypes generally, I'm interested to know how many of the credible NIMs include eQTL and whether that seems like a plausible mechanism of action for any of them.

---

## [Author Response]

Reviewing Editor comments:I enjoyed this paper. I include below a comment on top of those provided by the reviewers.In addition to matching to modern human SNPs for LD and MAF, it would be good to match on the basis of B-value (or a related metric). Loci with still polymorphic Neanderthal alleles are likely to be further from genes and in higher recombination regions. Doing this would allow us to move somewhat towards the conclusion that particular traits were under selection against Neanderthal alleles and not just selection against functional changes in general. The latter would still be of interest, but it's good to think about layered null models.

This is indeed an interesting question that could be answered by matching NIMs to modern human SNPs by matching on B-value in addition to LD and MAF. We attempted to estimate NIM heritability (*h*^2^*_NIM_*) and the difference in average heritability at NIMs to MH SNPs (Δh2) while matching on quintiles of MAF and LD bins, quartiles of B-value bins, and the ancestry status (NIM vs MH). A challenge with this approach is that attempting to jointly match on all of these metrics results in a large number of annotations with few NIMs in each annotation (this is not a problem with the MH SNPs which are about 30 times more numerous than the NIMs) so that *h*^2^*_NIM_* estimates are substantially less precise (estimated with standard errors that are about ten times larger on average than in the setting where we do not match on B-values) in this setting. Consequently, we do not find a significant difference in the per-SNP heritability at NIMs compared to MH SNPs.

To further investigate this question, we estimated (Δh2) matching on quartiles of B-values and quintiles of MAF. We now observe that the *h*^2^*_NIM_* estimates have standard errors that are comparable to the setting where we do not match on B-values. In this setting, we continue to observe a significant depletion in NIM heritability across phenotypes (53 phenotypes with Z-score < -3) with no evidence for traits with elevated (Figure 3—figure supplement 1). We have added these analyses to the main text (Section: The contribution of Neanderthal introgressed variants to trait heritability).

Taken together, our analyses suggest that depletion in heritability likely reflects selection against Neanderthal alleles rather than selection against variation in functionally constrained regions of the genome in general.

Reviewer #1 (Recommendations for the authors):In the section describing the whole-genome simulations of traits, it would help to clarify how the effect sizes are drawn. In particular, are the β values independent of frequency, but the per-SNP effects are normalized such that h^2 does not vary with frequency? This has been done in some previous studies for statistical convenience, and it has been argued that this type of model is qualitatively consistent with weak negative selection. In my opinion it is not a particularly plausible model for an effect size distribution, and when we have tried simulating negative selection models the joint distribution of effect sizes and frequencies can deviate widely from such procedures. Still, so long as the procedure is clarified I think it is fine for the purposes of this study.

In the section on the whole-genome simulations, we now clarify that each of the β values are drawn independently from a standard normal distribution (so it is independent of frequency) and are applied to standardized genotypes. We also further note in this section that:

“We note that when the causal SNPs are selected at random, this is the GCTA model that has been used in genetic studies of complex traits (Yang 2010).”

In general, I'm of the opinion that the best way to assess a method such as yours would be to perform forward in time simulations that include selection and demography and to assess the method on simulated traits, such that the joint distributions of frequencies/LD/effects is determined by the model. However, I recognize that this would be a large project and that you have made substantial effort to assess the method rigorously already. I also agree with what you said at the end of your paper, that the space of models that could correspond with your results is still quite vast, meaning that such simulations would necessarily focus on a small subset of potential models. I do think that adding some simulations with more extreme contributions due to rare variants would be helpful to establish the applicability of the method here.

We agree and have added new simulations, ULTRA RARE, in which SNPs with MAF < 0.01 constitute 90% of the causal variants (see details in response to main comment and Response Figure 2). We also strongly feel that the idea of running forward-in-time simulations with an explicit dependence on demographic and selection is valuable to test our method specifically and to systematically evaluate findings from similar studies in the field more generally. While we initially undertook such an effort as part of this study, we felt that the complexity of the model building warrants careful treatment that would be best achieved in a stand-alone study.

Reviewer #2 (Recommendations for the authors):I have 2 suggestions for analyses that would make this study stronger:1) This work makes an important point that future studies of introgressed haplotypes should account more carefully about MAF and LD patterns in their study populations of interest. Another choice such studies have to make is which set of putative introgressed haplotypes to use; there are several choices available that use LD information in different ways. It would be useful to know how robust these patterns are to differences in regions identified by those methods. It would also be helpful for interpreting differences between this and previous heritability studies; both McArthur 2021 and the Koller 2021 preprint cited in the appendix rely on the Sprime method of identifying introgressed haplotypes (Browning et al. 2018) rather than the CRF-based method used here. It would be informative to know at least how many haplotypes considered here are identified by the other methods, and whether they have the same distribution of MAF and LD score, and same general heritability trends.

We performed additional analyses with NIMs annotated by Sprime that were used in McArthur et al. 2021 (Source:https://github.com/emcarthur/trait-h2-neanderthals/tree/master/data/input_neanderthal_ regions). We used two sets of NIMs identified using Sprime in the 1000 Genomes Project and analyzed in McArthur et al. 2021: the least stringent set of 900,902 putatively introgressed variants identified in 1KG subpopulations regardless of evidence of matching the Neanderthal allele and the most stringent set of 138,774 putatively introgressed variants identified in the 1KG European subpopulations matching the Altai Neanderthal allele. We annotated the ancestry of the QC-ed SNPs in UKBB with the two sets of NIMs identified from Sprime and analyzed the heritability of the 96 UKB phenotypes at these NIMs (accounting for MAF and LD).

Of the set of 235,592 NIMs that were identified using the CRF, 73,675 NIMs are present in the most stringent set and 166,756 are present in the least stringent set. Consistent with our previous results from using NIMs annotated by CRF, we observe depletion in NIM heritability across traits. We also observe no enrichment in NIM heritability compared to MH SNPs suggesting that the discrepancy between our study and the McArthur et al. are methodological differences rather than due to using different NIM annotations. Appendix 5 Figure 3 shows the results for the least stringent set of NIMs analyzed in McArthur et al. 2021. The results from the most stringent set are added as Appendix 5 – Figure 4. We describe these results in Appendix 5.

We have also added figures to demonstrate that the distribution of MAF and LD score of NIMs annotated by Sprime are consistent with NIMs annotated using CRF (Appendix 5 – Figure 1 for the least stringent set with Appendix 5 – Figure 2 for the most stringent set). Overall, the NIMs annotated with Sprime show the same trend of MAF, LD score as NIMs annotated with CRF (as in our Figure 2). This suggests that the properties of NIMs reported in our paper are robust to the methodology used to identify them.

2) Focusing on coding changes is a reasonable starting point given their likelihood of affecting proteins. However, given the prevalence of noncoding changes associated with phenotypes generally, I'm interested to know how many of the credible NIMs include eQTL and whether that seems like a plausible mechanism of action for any of them.

Following this suggestion, we investigated the prevalence of eQTLs among credible NIMs. We used FUMA to annotate whether a credible NIM is an eQTL using eQTLs identified across the 54 tissues analyzed in GTEx v8. We find that 60.8% of the credible NIMs are eQTLs for at least one gene in at least one tissue. In contrast, only 25.6% of NIMs are eQTLs.

We also find that, out of 112 credible NIM sets, 23 have at least one credible NIM that alters coding sequence (as shown in our Figure 6), 79 have at least one credible NIM that works as an eQTL in at least one tissue while 22 have at least one credible NIM that impacts both coding sequence and gene expression.

Additionally, we listed the GTEx v8 tissues where credible NIMs are found to be eQTLs for genes expressed in those tissues (Figure 6—figure supplement 1). We find examples of credible NIMs for specific phenotypes that are eQTLs in relevant tissues. We find examples of credible NIMs for specific phenotypes that are eQTLs in relevant tissues: the credible set for measures of lung capacity (Forced Expiratory Volume (FEV1) and Forced Vital Capacity (FVC)) contains eQTLs for gene expressed in lung while credible sets for a measure of liver function (alanine aminotransferase levels in blood) contain eQTLs in liver. We have added these results to the main text in the Section: “Examination of the functional impact of credible NIMs”.